# Antiviral Activity of Eugenol Against Largemouth Bass Ranavirus Through Regulation of Autophagy and Apoptosis In Vitro and In Vivo

**DOI:** 10.3390/microorganisms13102281

**Published:** 2025-09-30

**Authors:** Yewen Wang, Lifang Cao, Leshan Ruan, Xingyu Chen, Chunhui Song, Shina Wei, Yunchang Xie

**Affiliations:** 1Key Laboratory of Biodiversity Conservation and Bioresource Utilization of Jiangxi Province, College of Life Sciences, Jiangxi Normal University, Nanchang 330022, China; wangyewen@stu.scau.edu.cn (Y.W.); 15279862930@163.com (X.C.); schxzx1977@126.com (C.S.); 2College of Marine Sciences, South China Agricultural University, Guangzhou 510642, China; 18948120826@163.com (L.C.); ruanleshan@126.com (L.R.)

**Keywords:** largemouth bass, eugenol, antiviral, autophagy, apoptosis

## Abstract

Largemouth bass ranavirus (LMBV) causes high mortality rate in largemouth bass during outbreaks, resulting in huge economic losses. Eugenol (EUG) has potent antiviral activity, showing promising potential against LMBV. Thus, to investigate EUG’s efficacy against LMBV, corresponding analysis was conducted in vivo and in vitro. Firstly, EUG demonstrated to be able to down-regulate both the mRNA and protein levels of the major capsid protein (MCP) in LMBV-infected cells. In addition, EUG could inhibit the expression of cleaved-caspase-3 in LMBV-infected fathead minnow (FHM) cell. On the other hand, EUG would not only directly regulate the protein kinase B (AKT)/mammalian target of rapamycin (mTOR) pathway but also affect the AMP-activated protein kinase (AMPK) pathway in FHM cells during LMBV infection. These results indicated that EUG exerts its antiviral effects by modulating both LMBV-induced apoptosis and autophagy. Notably, EUG reduced the viral load present within the tissues of LMBV-infected largemouth bass, thereby ultimately enhancing their survival rate in the culture environment by about 20%. These mechanistic assays revealed the anti-LMBV properties of EUG, which could significantly enrich the research content of plant extracts in the field of aquatic antiviral, and provide important theoretical basis for the development and application of related products.

## 1. Introduction

Largemouth bass ranavirus (LMBV), with an icosahedral shell enveloping a double-stranded DNA, was originally discovered from largemouth bass (*Micropterus salmoides*) in the United States [1,2]. Meanwhile, largemouth bass has gradually become an important economic fish and is widely cultured in China due to its tasty meat, but this aquaculture industry is continuously damaged by the threat of LMBV, one of the most harmful pathogens [3,4]. Although LMBV exhibits a broad host range, natural infections primarily affect largemouth bass [5]. LMBV is most prevalent in summer, with outbreaks more likely at 25–30 °C. The symptoms of LMBV infection are very different by region. In China, the most common clinical symptoms of LMBV infection in bass are characterized by the swelling of internal organs, body surface ulceration, and muscle necrosis [6,7]. Previous study revealed that LMBV infection can significantly trigger immune activation and inflammatory response in largemouth bass [6]. Inhibiting largemouth bass nuclear factor kappa B (NF-κB)/P65 in vivo significantly promoted LMBV infectivity, while overexpression of *Micropterus salmoides* P65 in vitro positively regulated interferon and related genes, exerting antiviral effects [8]. LMBV infection induces distinct cell death pathways depending on the host cell type. Specifically, it could promote apoptosis in fathead minnow (FHM) cells and autophagy couple with apoptosis in epithelioma papulosum cyprinid (EPC) cells [9]. In contrast, it caused non-apoptotic cell death in *M. salmoides* fin (MsF) cells [10]. Inhibitors of phosphatidylinositol 3-kinase (PI3K) pathway and extracellular-signal-regulated kinases (ERK) signaling pathways in EPC cells significantly suppress LMBV replication and regulate LMBV-induced apoptosis [11]. Furthermore, promoting cellular autophagy has been shown to confer antiviral effects. Collectively, these findings highlight autophagy and apoptosis as two key pathological mechanisms underlying LMBV infection and pathogenesis.

Eugenol (EUG) is a major volatile component of clove essential oil and possesses broad antiviral properties that have been shown to work in many studies [12,13]. It has been shown to attenuate herpes virus-induced keratitis and inhibit the viral infection in mice [14]. In previous work, we have verified that EUG inhibits the expression of inflammatory factors caused by Singapore grouper iridovirus (SGIV) in grouper spleen (GS) cells while alleviating cell oxidative stress through reduction of reactive oxygen species (ROS) levels [15]. Additionally, it has been reported that EUG inhibited influenza A virus (IAV) infection by suppressing the activation of mitogen-activated protein kinase (MAPK) and inhibitor of kappa B kinase (IKK)/NF-kB pathway, which is consistent with our previous findings regarding its inhibitory effect on activation of the SGIV-induced MAPK pathway [16].

A growing number of studies are increasingly focused on investigating the potential role of EUG in modulating autophagy and apoptosis across diverse physiological and biochemical contexts. In breast cancer cells, EUG induced autophagy by up-regulating the threonine kinase 1- forkhead box O3 (AKT–FOX3) pathway to promote apoptosis, and at the same time, up-regulating the expression of microtubule-associated protein 1 light chain 3 (LC3I) and down-regulating the expression of nucleoporin 62 (NU P62) [17]. EUG has also been shown to attenuate the inflammatory response by inhibiting the PI3K/AKT pathway in studies using a mouse model of conjunctivitis [18]. Furthermore, recent studies also indicate that high-concentration EUG treatment modulates the AMPK/AKT/mTOR pathway, thereby attenuating acrylamide (ACR)-induced testicular toxicity in mammals [19], suggesting that EUG can affect AKT-related signaling pathways in different diseases. Viral infections are closely associated with various forms of cell death, including apoptosis, necrosis, autophagy, pyroptosis, and ferroptosis. It is commonly believed that infection leads to cell death and an inflammatory response, but it has also been shown that altruistic cell death of infected cells can be beneficial to the organism as a whole, ultimately limiting the path of virus production [20]. Autophagy has been implicated in the development of a lot of viral diseases. Nevertheless, the relationship between EUG and LMBV infection remains poorly understood, highlighting a significant knowledge gap and the need for further investigation.

In this study, we investigated the effectiveness of EUG against LMBV and elucidated the possible cellular pathways such as autophagy and apoptosis. Here we found that 75 μM EUG group markedly reduced the mRNA expression of the LMBV–MCP genes. Our results show that EUG can reduce the apoptosis induced by LMBV in FHM cells and enhance autophagy in abovementioned cells by regulating AKT/mTOR and AMPK signaling pathways. Moreover, EUG could potently suppress the transcriptional levels of LMBV genes and improve survival outcomes in LMBV-challenged largemouth bass. These antiviral effects obviously revealed that EUG could be a potential drug for the prevention and treatment of LMBV, offering practical value to the aquaculture industry.

## 2. Materials and Methods

### 2.1. Ethics Statement

Our experiments with laboratory animals adhered to the South China Agricultural University’s (SCAU) institutional ethical guidelines. This study was approved by SCAU (ethical protocol code: 2020G009), and a certificate of approval is available upon request.

### 2.2. Fish, Cells and Virus

Two hundred largemouth bass (8.0 ± 2 cm) were purchased from a fish farm in Guangdong Province and prepared for antiviral and toxicity analyses. The fish were cultured in a recirculating water system at 25–30 °C and fed for a 10-day acclimation period before being used in the experiments. The following water quality parameters were maintained: dissolved oxygen > 5 mg/L, pH 7.5 ± 0.2, and ammonia nitrogen at approximately 1.0 mg/L. Prior to the formal experiment, 10 fish were randomly selected for PCR detection to ensure that they were LMBV negative. LMBV was isolated from diseased largemouth bass and stored in our laboratory [11].

The FHM cell line was propagated according to the recommended methods using M199 culture medium (Gibco, Waltham, MA, USA) supplemented with 10% fetal bovine serum (Gibco, Waltham, MA, USA) at 28 °C [21].

### 2.3. Cell Treatment

EUG (Sigma-Aldrich, St. Louis, MO, USA, E51791) was diluted to 500,000 μM in dimethyl sulfoxide (DMSO) at the time of use and further diluted to working concentrations in M199 medium containing 10% FBS (Gibco, Waltham, MA, USA). For viral infection experiments, cells were treated with different concentrations of EUG according to the experimental requirements, and cell cultures were pretreated with varying concentrations of EUG for 2 h, followed by LMBV infection at a multiplicity of infection (MOI) of 1 (MOI = 1). The control group was treated with an equal volume of DMSO without the addition of EUG, and the final volume concentration of DMSO in the culture medium was adjusted to 0.1% uniformly.

### 2.4. Cytotoxicity Assays

The cytotoxic effects of EUG on FHM cells were evaluated using the Cell Counting Kit-8 (US Everbright @Inc., Suzhou, China) assay in 96-well plates. Five groups of EUG concentrations were tested: 0 μM (DMSO), 50 μM, 75 μM, 100 μM, and 150 μM, with 8 replicates in each set. Cells were seeded in 96-well plates at a density of 1.0 × 10^4^ cells per well and incubated at 28 °C for 24 h. When cell confluence reached 80–90%, cells were treated with EUG or DMSO for 24 h. Cell viability was assessed with a microplate reader (Thermo Fisher, Waltham, MA, USA) at 450 nm after 2 h incubation in serum-free medium with Cell Counting Kit-8 and protection from light.

### 2.5. Virus Infection Assay In Vitro

FHM cells were cultured in 24-well or 6-well plates and treated with EUG or DMSO for 2 h prior to LMBV infection. Cells were collected at different time points for subsequent experiments, including qRT-PCR and western blotting. Specifically, after 12 or 24 h of viral infection, the medium was removed. The cells were gently washed with 1 mL of pre-chilled PBS per well to maintain cellular integrity, incubated on ice for 2 min, and then the liquid was completely collected.

### 2.6. RNA Isolation and qRT-PCR Analysis

Tissues were ground in pre-cooled PBS with a homogenizer (Beijing Solarbio Science, Beijing, China). Total RNA was isolated using the Animal/Cell Total RNA Isolation Kit (FOREGENE, Chengdu, China) and reverse-transcribed with the ReverTra Ace qPCR RT Kit (TOYOBO, Osaka, Japan) [22]. At the indicated time points, LMBV-infected cells were harvested for RNA extraction as described previously [15]. In brief, the total RNA of infected cells was extracted using a cell total RNA isolation kit (FOREGENE, Chengdu, China) according to the manufacturer’s protocol. The RNA was reverse-transcribed using a ReverTra Ace-a kit (Toyobo, Osaka, Japan). The transcription level of viral genes, including MCP and myristoylated membrane protein (MMP), were examined using qRT-PCR analysis. qRT-PCR was performed using SYBR^®^ Green Realtime PCR Master Mix (Toyobo, Osaka, Japan) in an Applied Biosystems QuantStudio 5 Real Time detection system (Thermo Fisher Scientific, Waltham, MA, USA) as described previously. The cycling conditions for qRT-PCR were as follows: 95 °C for 5 min for activation followed by 45 cycles at 95 °C for 5 s, 60 °C for 10 s, and 72 °C for 15 s. The primers utilized for the PCR are detailed in Table 1. Moreover, the primers used in this experiment had annealing temperatures ranging approximately between 60 and 65 °C and were validated by melt curve analysis and standard curve quantification, demonstrating amplification efficiencies of 96-100% with R^2^ > 0.96. The 2^−ΔΔCT^ method was used to calculate the relative expression ratio of the selected genes to β-actin (reference gene). Statistics were calculated using SPSS version 20. Differences were considered to be statistically significant when * *p* < 0.05.

### 2.7. Western Blotting Analysis

Treated FHM cells were collected and subjected to the western blotting assay according to the methods and materials previously used [23]. FHM cells were first digested with trypsin and collected on ice and lysed in Pierce IP lysis buffer (Thermo, Waltham, MA, USA) for 30 min, followed by centrifugation at 12,000× *g* for 10 min to remove cellular debris, and the supernatant was mixed with 5 × protein sampling buffer and boiled for 10 min. Proteins were separated by 10% sodium dodecyl sulfate-polyacrylamide gel electrophoresis and transferred to 0.22 μm Immobilon polyvinylidene difluoride membranes (Millipore, Temecula, CA, USA), and the blots were incubated in 5% skimmed milk. The blots were then incubated with specific primary antibodies for 2 h. After three washes with phosphate-buffered saline-Tween, the blot was incubated with peroxidase-conjugated secondary antibody IgG (1:5000 dilution, Abcam, Cambridge, UK, ab172730). Enhanced chemiluminescence (Thermo, Waltham, MA, USA) was used to visualize the immunoreactive bands. The specific primary antibodies used in the experiments were as follows: LMBV major capsular protein (MCP) (1:2000 dilution, GENECREATE, Wuhan, China), cleaved caspase3 (1:1000 dilution, CST, Kansas City, MO, USA, ab32042), phospho-AKT (1:2000 dilution, CST, ab279732), phosphor-mTOR (1:1000 dilution, CST, Kansas City, MO, USA, ab211061), Beclin1 (1:1000 dilution, Proteintech, Rosemont, IL, USA, ab207612), LC3 (1:2000 dilution, Abcam, Cambridge, UK, ab63817), P62 (1:1000 dilution, Abcam, Cambridge, UK, ab207305), autophagy-related protein 5 gene (ATG5) (1:1000 dilution, Abmart, Cambridge, UK, T55766S), phospho-AMPK (1:1000 dilution, CST, Kansas City, MO, USA, ab68206), and β-tubulin (1:2000 dilution, Abcam, Cambridge, UK, ab6046). Quantification was performed by ImageJ 1.51 software (NIH, Bethesda, MD, USA) and standardized by β-tublin.

### 2.8. Annexin V-FITC/PI Apoptosis Assay

Cells were seeded in 24-well plates at a density of 5.0 × 10^4^ cells per well and incubated at 28 °C for 24 h. When cell confluence reached 80–90%, FHM cells were infected with LMBV after treatment with EUG or DMSO for 2 h. Cells were collected by trypsinization into 1.5 mL EP tubes (Axygen, Union City, CA, USA) and washed twice with cold PBS (0.15 mol/L, pH 7.2). Cells were centrifuged at 3000 rpm for 5 min, after which the supernatant was discarded and the precipitate was resuspended in 1× binding buffer (Thermo Fisher Scientific, Waltham, MA, USA). A total of 100 μL sample solution was transferred to 5 mL culture tubes and incubated with 5 μL FITC-labeled Annexin V (Pharmingen) and 5 μL PI (Pharmingen) for 15 min at room temperature in the dark. A total of 400 μL of 1× binding buffer was added to each sample tube and then analyzed by flow cytometry. Each sample was measured in triplicate. The obtained data were analyzed using FlowJo 10.4 software, and cell populations were differentiated via Annexin V-FITC/PI dual-parameter gating into three distinct groups as follows: viable (Annexin V^−^/PI^−^), early apoptotic (Annexin V^+^/PI^−^), and late apoptotic/necrotic (Annexin V^+^/PI^+^) cells, with proportional data extracted for statistical comparisons. The results presented were from one representative experiment.

### 2.9. Antiviral Activity of EUG In Vivo

We tested the safe concentration of EUG in fish injected intraperitoneally and then conducted anti-LMBV experiments on subgroups of largemouth bass. In the EUG safety concentration experiment, juvenile largemouth bass were divided into 5 groups of 20 individuals per group and injected with 100 μL of PBS containing different concentrations of EUG (0 mg/kg, 10 mg/kg, 20 mg/kg, 50 mg/kg, and 100 mg/kg) for one week of observation. The mortality rate was recorded daily. After one week, the intestines, spleens, and livers were removed and tissue sections were prepared to observe pathological changes induced by EUG. For the antiviral assay, juvenile largemouth bass were divided into the following two groups: a control group and an experimental group that was injected with LMBV + EUG (10 mg/kg). The control group was injected with 100 μL of LMBV at a titer of 10^6^ TCID_50_/mL in PBS, while the experimental group received 100 μL of a mixture of LMBV and EUG. Mortality rates were recorded daily over a period of 7–14 days. The intestines, spleens, and livers were then collected and tested for LMBV viral load in the tissue by qRT-PCR.

### 2.10. Statistical Analysis

All data were from at least three independent replicated experiments, results are expressed as mean ± standard deviation (SD), and the data were analyzed using GraphPad Prism 9.5. Statistical significance was assessed using SPSS version 20 by Student’s t-test and one-way/two-way analysis of variance (ANOVA) with 95% confidence intervals. Significance was set at * *p* < 0.05.

## 3. Results

### 3.1. Effect of EUG in Different Concentrations on LMBV Infection

FHM cells were initially selected to establish an in vitro model to study the effect of EUG on LMBV infection. FHM cells were seeded in 96-well plates at a density of 1.0 × 10^4^ cells per well and incubated at 28 °C for 24 h. At 80–90% confluence, the cells were treated with EUG or DMSO for 24 h, followed by cell viability measurement using the CCK-8 assay. Based on the results, we determined that the maximum safe concentration of EUG in FHM cells was 75 μM, whereas treatment with 100 μM EUG significantly suppressed cellular activity, reducing viability to approximately 90% (Figure 1a). The major capsid protein (MCP) and myristoylated membrane protein (MMP) of iridovirus play pivotal roles in the viral life cycle, participating in critical processes including host cell recognition and viral entry [24,25]. Given their essential functions in viral pathogenesis, the quantitative changes in MCP and MMP expression levels were employed as key virological markers to assess viral replication efficiency in this study. We selected 50 μM and 75 μM EUG to pre-treat FHM cells before LMBV infection for 2 h. At 24 h after infection, only the 75 μM EUG group markedly reduced the mRNA expression of the LMBV–MCP and LMBV–MMP genes (by 1.5-fold and 0.2-fold, respectively) (Figure 1b). Therefore, 75 μM EUG was used as the optimal treatment for FHM cells in all subsequent experiments.

### 3.2. EUG Inhibits LMBV Infection at Different Time Points

To determine the in vitro antiviral efficacy of EUG, cells were incubated with 75 μM EUG for 2 h and then infected with LMBV. The mRNA and protein levels of LMBV genes were examined at different time points by qRT-PCR and western blotting. Viral gene mRNA levels were significantly reduced in EUG-treated groups at various time points (Figure 2a). Additionally, LMBV–MCP protein synthesis was also significantly reduced, indicating that the EUG was capable of inhibiting LMBV infection of FHM cells (Figure 2b,c).

### 3.3. EUG Inhibits Apoptosis in Cells Infected by LMBV

As demonstrated by earlier research, LMBV infection instigates a process of apoptosis in FHM cells, whilst apoptosis concurrently serves to promote LMBV virus replication [11]. In order to ascertain the antiviral mechanism of EUG, we explored its effect on LMBV-induced apoptosis. FHM cells were exposed to EUG for a duration of 2 h prior to infection with LMBV or left untreated. Flow cytometry and western blotting were conducted to ascertain the degree of apoptosis. The analysis of cell samples by flow cytometry demonstrated that EUG had no influence on the degree of apoptosis in healthy cells; infection for 12 h significantly promoted FHM cell apoptosis, whereas EUG can inhibit corresponding FHM cell apoptosis induced by LMBV (Figure 3a,b). Subsequently, cleaved caspase-3 protein, an apoptosis marker in FHM cells, was detected by western blotting. The results indicated that its levels were significantly lower in the EUG-treated group than in the control group, with the level in the EUG-treated group being approximately half (50%) of that in the control group (Figure 3c). Overall, EUG significantly inhibited LMBV-induced FHM cell apoptosis.

### 3.4. EUG Promotes Cellular Autophagy in LMBV Infection

LMBV infection leads to autophagy in EPC cells, and further promotion of autophagy inhibits LMBV replication and cell apoptosis [9]. It is known that the detection of Beclin1, LC3, and P62 proteins are key indices to evaluate autophagy level. We determined the level of autophagy in FHM cells by western blotting to investigate the mechanism of inhibition of apoptosis by EUG. FHM cells were harvested after EUG treatment and infection with LMBV for 12 h. The results in Figure 4 indicate that the levels of autophagy-related proteins (LC3I/II, Beclin-1, and ATG5) were significantly increased in the EUG-treated group compared with the control group, while the level of P62 protein was decreased, suggesting that EUG treatment could increase the autophagy level of LMBV-infected cells.

### 3.5. EUG Regulates AMPK and AKT/mTOR Pathway in LMBV Infection

Autophagy, a highly conserved cellular degradation process, is primarily regulated through several key signaling pathways, with the AKT/mTOR and AMPK pathways playing central roles in its modulation. The AKT/mTOR signaling axis serves as a critical negative regulator of autophagy. To explore the signaling pathway through which EUG regulated autophagy, the protein levels of p-AMPK, p-mTOR, and p-AKT were assessed by western blotting. As presented in Figure 5, EUG treatment significantly reduced the phosphorylation of AKT by 2.6-fold and that of mTOR by 0.2-fold, while increasing AMPK phosphorylation by 0.8-fold. Therefore, AMPK and AKT/mTOR pathway participated in the autophagy mechanisms of EUG in LMBV infection.

### 3.6. EUG Reduces Mortality in LMBV-Infected Largemouth Bass

To determine the in vivo toxicity of the drug, EUG was diluted to different concentrations and injected intraperitoneally into largemouth bass and observed for 7 days. Intraperitoneal injection of 10 mg/kg of EUG did not result in any mortality (Figure 6a). Histopathological results showed that there were no significant differences in the intestines, spleens, and livers between the 10 mg/kg EUG injection group and the control group (Figure 6b). To explore the in vivo effect of EUG on LMBV infection, largemouth bass were intraperitoneally injected with a mixture of 10 mg/kg EUG and LMBV. It was found that the survival rate of the mixed EUG injection group reached 80%, which was significantly different from the 60% survival rate of the control group after 12 days (Figure 6c). The death caused by LMBV infection was inhibited by injection of EUG in vivo.

### 3.7. EUG Inhibits LMBV Infection in Tissues

Several LMBV-infected largemouth bass tissues were collected and the mRNA levels of viral genes were tested by qRT-PCR after 12 days. The results showed that the mRNA levels of LMBV–MCP and LMBV–MMP in the intestines, spleens, and livers of the EUG injected group were significantly lower than those of the control group, which confirms that EUG can inhibit the LMBV infection in vivo (Figure 7a–c).

## 4. Discussion

Largemouth bass is widely distributed throughout the freshwater waters of the United States and Canada [23]. It was later brought to Asia and is currently the largest farmed freshwater fish in China, with higher economic and environmental benefits [26]. As the scale of aquaculture continues to expand, the impact of disease on largemouth bass aquaculture becomes apparent. Due to the lack of effective treatment, viral diseases are resulting in serious economic losses. Iridovirus is a large, structurally complex, 20-sided, and double-stranded DNA virus [27]. This virus infects fish, reptiles, and amphibians, causing severe disease and varying degrees of mortality [27]. LMBV is a member of the Iridoviridae family, specifically the genus Frog Virus [28]. Infection of LMBV can result in a variety of clinical symptoms, including ulceration of the body surface, muscle necrosis, swelling of internal organs, etc., which are frequently accompanied by high mortality during the epidemic season. In addition, pathological analysis revealed hepatocellular nuclear consolidation and extensive necrosis of the spleen with lymphopenia [29]. What is more serious is that there are currently no approved drugs or vaccines available to control LMBV.

Faced with the aforementioned crisis, exploring antiviral ingredients from plant extracts has gradually become an effective solution and strategy [7,30]. In a previous study, we have shown that EUG can inhibit SGIV infection in vitro, which together with LMBV are members of the iridoviridae family [15]. EUG has a well-established antiviral effect in vitro. To learn more about its antiviral mechanism and potential applications in aquaculture, we looked into how it affected LMBV infection both in vivo and in vitro. We chose FHM cells as in vitro infection model, investigated the optimal antiviral concentration and antiviral effect of EUG in cells, while detecting the infection situation of LMBV in cells by analyzing the expression levels of MCP and MMP [31]. Among them, MCP is the major structural protein of iridoviruses and has low homology in the iridovirus family, which is widely used for virus detection [24,32]. MMP serves as the viral envelope protein in iridoviruses, playing key roles in host cell recognition, viral assembly, and entry [25]. According to the findings of the cytotoxicity assessment of EUG, the viability of FHM cells remained unaffected at concentrations below 100 μM. Furthermore, we treated FHM cells with different concentrations of EUG (50 μM, 75 μM) for 2 h before being infected with LMBV, and discovered that 75 μM EUG could significantly reduce the expression of LMBV–MCP and LMBV–MMP (by 1.5-fold and 0.2-fold, respectively). Thus, it was suggested that 75 μM of EUG significantly inhibited LMBV infection of FHM cells.

Apoptosis is a kind of programmed cell death. In contrast to cell necrosis, apoptosis is closely related to cell proliferation in healthy state [33]. In the process of defense against viral infection, cells can inhibit viral replication by rapidly activating apoptotic pathways to make infected cells die prematurely. Conversely, viruses can manipulate apoptosis to promote their own replication and spread. The protein encoded by human immunodeficiency virus-1 (HIV-1) depletes T cells by making them sensitive to apoptosis through different regulatory mechanisms [34]. This mechanism is also present in RNA virus infection, such as West Nile virus (WNV) and Zika virus (ZIKV) infections, which can induce neurotoxicity by mediating nerve cell apoptosis through various pathways [35,36]. Many fish viruses cause different types of cell death in a cell-dependent manner. SGIV causes non-apoptotic cell death in GS cells, but apoptosis in FHM cells [37]. According to earlier research, LMBV infection was found to induce apoptosis in FHM and EPC cells and non-apoptotic cell death in MsF cells [10]. In addition, EUG has been demonstrated in many previous cancer studies to promote apoptosis and cancer cell toxicity. Low dose of EUG can induce apoptosis of different breast cancer cells and inhibit the proliferation of cancer cells. Kim et al. demonstrated that EUG has the potential to induce apoptosis of melanoma and neuroblastoma cells [38,39]. Thus, in order to further investigate the mechanism of EUG inhibiting LMBV infection, we employed flow cytometry and western blotting to analyze the effect of EUG on LMBV-induced apoptosis. The following results revealed EUG had no influence on the apoptosis in healthy cells but can inhibit LMBV infected FHM cell apoptosis. The initiation and execution of apoptosis are regulated by Bcl-2 and Caspase family proteins. Caspase-3 is a crucial protease for apoptosis and cleaved-caspase-3 is the primary executor of apoptosis [40]. In our study, we observed the negative effect of EUG on apoptosis by detecting it can directly inhibit the expression of cleaved-caspase-3 in LMBV infected FHM cell. Therefore, abovementioned results suggest that EUG can considerably prevent apoptosis induced by LMBV infection. In the present study, all conclusions on apoptosis are strictly based on data from the experimental models and conditions applied in this study, with no extrapolation beyond these specific contexts. Moreover, EUG significantly inhibited LMBV-induced apoptosis in FHM cells during 12–24 h treatment, demonstrating clear short-term anti-aprototic efficacy. However, to further elucidate its long-term effects and pharmacological mechanisms, future work will extend treatment duration (e.g., 36–72 h) and systematically examine time-dependent responses in apoptosis-related indicators. This approach will enhance understanding of EUG’s anti-apoptotic mechanisms and support its potential application in aquatic antiviral therapy and cytoprotection.

Autophagy, which is closely related to viral infections, plays a crucial role in the immune system. It functions as an essential defense mechanism against autoimmunity and exhibits antiviral properties. Conversely, certain viruses can exploit autophagy to facilitate viral replication and enhance infection [41,42]. For instance, autophagy can limit the replication of the virus in cells while HIV inhibits autophagy during infection [43,44]. The interaction between apoptosis and autophagy plays an important role in body development and homeostasis, and the association between them has been reported in many different viruses [45,46,47,48]. In the study of IAV, inhibition of autophagy resulted in the blockage of IAV replication and limited IAV protein-induced apoptosis [48]. In the relationship between autophagy and apoptosis induced by Epstein-Barr virus, it was discovered that autophagy inhibited the release of virus by suppressing apoptosis [45]. ATG5 and LC3I/II play an important role in the occurrence and development of autophagosomes and autophagic vesicles, in which ATG5 is involved in each stage of the formation of autophagosomes. LC3I/II is the direct evidence to determine whether autophagy occurs [21,49]. The autophagy-related gene Beclin 1 was originally discovered in 1999 and is the mammalian homologue of ATG6 [50]. In mammalian cells, Beclin 1 can interact with Bcl-2 to regulate cellular autophagy. Overexpression of Beclin 1 may have anticancer and antiviral effects [47,51]. P62 is a ubiquitinated protein that can ubiquitinate autophagosomes upon binding to LC3; therefore, inhibition of autophagy leads to significant accumulation of P62, which is conversely depleted [52,53]. Therefore, we examined the protein expression levels of LC3I/II, ATG5, Beclin 1, and P62 to investigate the effect of EUG on LMBV-induced autophagy in FHM cells, which obviously indicated EUG can increase the expression of LC3I/II, Beclin-1, and ATG5 and decrease the expression of P62 protein. These results showed that EUG could significantly promote autophagy in LMBV-infected FHM cells. In this study, we preliminarily confirmed that EUG induces autophagy by examining autophagy marker proteins, such as LC3-II accumulation and p62 degradation. To further elucidate the promotive effect of EUG on autophagic flux, future studies will employ autophagic flux inhibitors (e.g., chloroquine CQ or bafilomycin A1) to explicitly block and monitor the completeness of the autophagic process, thereby verifying the mechanism by which EUG regulates autophagic flux.

AMPK is an evolutionarily conserved serine/threonine protein kinase involved in a wide range of cellular activities and can be phosphorylated under many physiological or pathological conditions [54]. Under starvation or some pathological conditions, activated AMPK can inhibit mTOR activity and indirectly activate FOXO3 targets thereby promoting autophagy [55]. mTOR is an important autophagy regulatory protein and regulates a variety of downstream transcription factors; many viral infections activate the mTOR pathway. For example, in HIV infection, activation of autophagy by rapamycin reduces virus-induced apoptosis by inhibiting mTORC1, which further inhibits the AKT/PKB pathway [56,57]. AKT is a key signal transduction protein. After activation of PI3K, AKT is recruited to the cell membrane and exposed to phosphorylation sites, and phosphorylated AKT is transferred to the cytoplasm or nucleus, indirectly promoting mTORC1 to exert the function of inhibiting autophagy [58,59]. In fact, the AKT pathway can be activated by many viral infections. *Polyomavirus* spp. enhances AKT signaling by attacking phosphatase activity thereby causing accelerated cellular accretion and division, allowing the cell to enter the S-phase, at which point viral DNA synthesis begins [60]. Our experimental results suggest that EUG inhibits AKT and mTOR phosphorylation and promotes AMPK phosphorylation in LMBV-infected FHM cells, which confirmed that AMPK and AKT/mTOR pathways are critical routes of EUG promoting cellular autophagy during LMBV infection. In the present experiment, the role of EUG through the AMPK and AKT/mTOR pathways was inferred solely based on phosphorylation levels, without the use of AMPK or AKT/mTOR pathway inhibitors or activators to validate its antiviral effects. However, directly perturbing the AMPK and AKT/mTOR pathways is crucial to verify whether EUG’s anti-LMBV effects are functionally mediated through these signaling mechanisms. In future studies, we will further elucidate the specific mechanism by which EUG modulates these pathways to exert anti-LMBV effects under conditions of AMPK and AKT/mTOR pathway intervention.

Finally, to verify the validity of the EUG in vivo experiments, we intraperitoneally injected a safe concentration of EUG mixed with a half-lethal concentration of LMBV into largemouth bass. The survival rate of fish in EUG treatment group increased by about 20% compared to the control group, and in subsequent assays, the mRNA expression of LMBV–MCP and LMBV–MMP genes within the tissues was much lower than that in the control group. Therefore, we conclude that EUG has significant anti-LMBV activity both in vivo and in vitro (Figure 8).

This study demonstrated the significant anti-LMBV efficacy of EUG via intraperitoneal injection, as evidenced by markedly improved survival rates and substantially reduced viral loads across multiple tissues in largemouth bass. Future research will prioritize the development of practical administration methods such as immersion and oral delivery, while comparing differences in viral load, immune response, and survival rates. Further work will focus on establishing dose–response relationships, optimizing delivery strategies, and evaluating long-term safety and tissue residues to meet regulatory requirements for commercial antiviral agents, ultimately advancing EUG toward becoming a commercially viable antiviral drug in aquaculture.

## 5. Conclusions

In summary, EUG can exert anti-LMBV effect in vivo and in vitro, mainly by inhibiting apoptosis and promoting autophagy in LMBV infected cells. We further explored the in vivo antiviral effect of EUG via intraperitoneal injection. The results showed that intraperitoneal injection of EUG could reduce the mortality of largemouth bass due to LMBV infection and inhibit the mRNA levels of LMBV–MCP and LMBV–MMP in tissues. These suggest EUG plays a defensive role in LMBV infection. This work positions EUG as a promising therapeutic candidate and provides a novel theoretical framework for its future application in antiviral strategies.

## Figures and Tables

**Figure 1 microorganisms-13-02281-f001:**
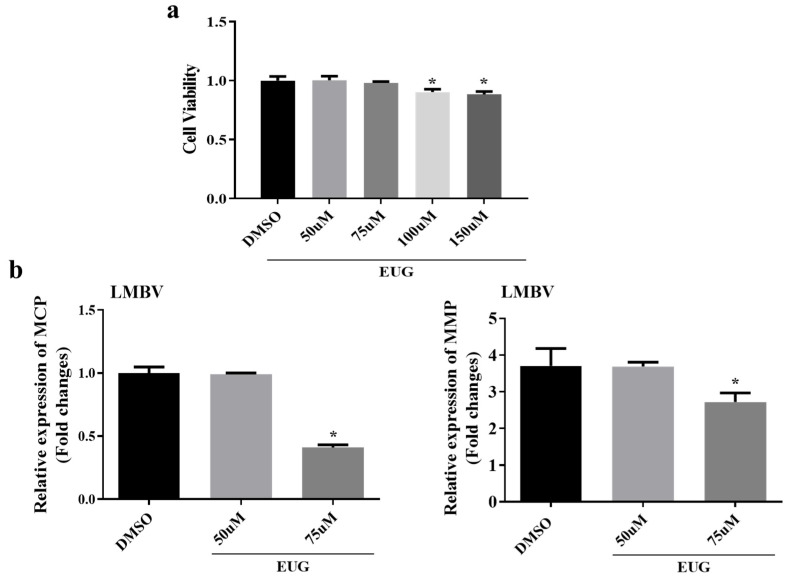
Maximum safe concentration of EUG and its effect on LMBV infection (MOI = 1). (**a**) Different concentrations of EUG (50 μM, 75 μM, 100 μM, and 150 μM) and DMSO (control) were co-cultured with FHM cells for 24 h. (**b**) After treating FHM cells with DMSO or different concentrations of EUG (50 μM, 75 μM) for 2 h, FHM cells were infected with LMBV, and the mRNA expression levels of LMBV–MCP and LMBV–MMP were detected by qRT-PCR after 24 h. Data expressed as mean ± SD of three independent experiments; each experiment was performed in triplicate. After confirming normal distribution, Student’s t-test was applied for analysis. * *p* < 0.05.

**Figure 2 microorganisms-13-02281-f002:**
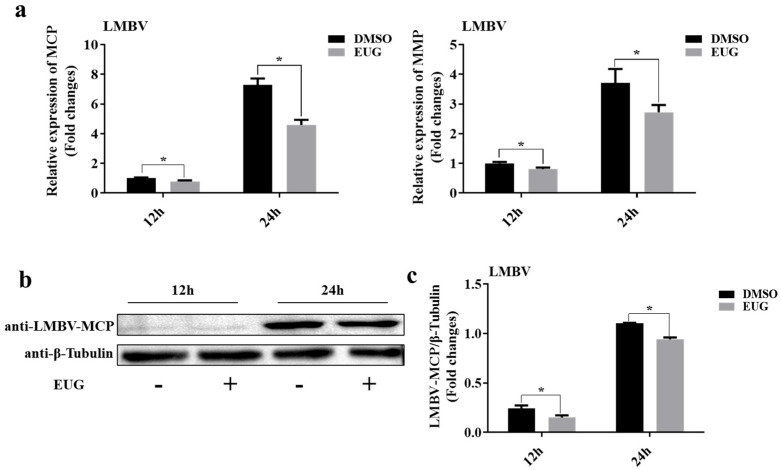
Effect of EUG on LMBV infection at different time points. (**a**) Cells were infected with LMBV after treatment with DMSO or EUG for 2 h. After 12 and 24 h, the gene expression levels of LMBV–MCP and LMBV–MMP were detected by qRT-PCR. (**b**) Cells were collected 12 and 24 h after infection, and LMBV–MCP protein and cellular β-tubulin were analyzed by western blotting. (**c**) MCP/β-Tubulin values were calculated for each group as described above. Data expressed as mean ± SD of three independent experiments; each experiment was performed in triplicate. After confirming normal distribution, two-way ANOVA was applied for analysis; * *p* < 0.05.

**Figure 3 microorganisms-13-02281-f003:**
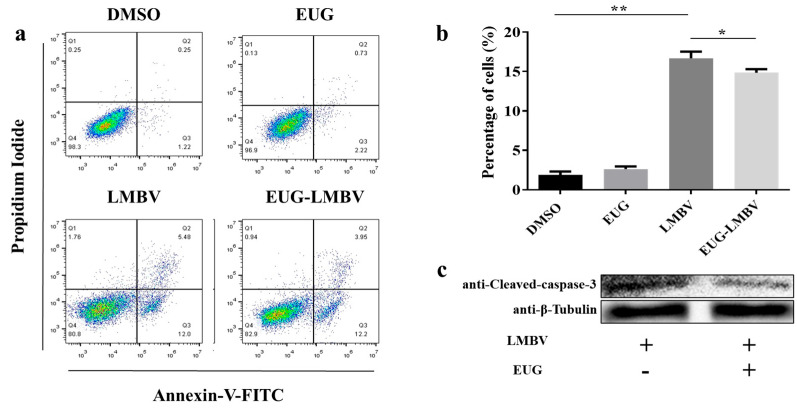
EUG inhibits apoptosis induced by LMBV infection. (**a**) Cells were incubated with EUG for 2 h before LMBV infection and apoptosis was detected by annexin V/PI double staining after 24 h. Q3 represents the percentage of total or early apoptotic cells and the differences between groups were analyzed. (**b**) Quantitative analysis of the percentage of apoptotic cells. (**c**) FHM cells were treated with EUG for 2 h and infected with or without LMBV, and samples were collected 12 h later to detect cleaved-caspase-3 protein and cellular β-tubulin. + indicates the addition of the virus/drug shown on the left, - indicates its absence. Data expressed as mean ± SD of three independent experiments; each experiment performed in triplicate. After confirming normal distribution, Student’s t-test and one-way ANOVA were applied for analysis; * *p* < 0.05; ** *p* < 0.01.

**Figure 4 microorganisms-13-02281-f004:**
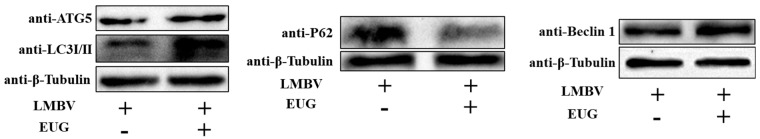
EUG promotes the accumulation of autophagy in FHM cells. FHM cells were infected with LMBV after incubation with EUG for 2 h. Cell samples were taken 24 h later and protein expression levels of the autophagy-related proteins LC3I/II, ATG5, P62, and Beclin1 were detected by western blotting. Data expressed as mean ± SD of three independent experiments; each experiment performed in triplicate. + indicates the addition of the virus/drug shown on the left, - indicates its absence.

**Figure 5 microorganisms-13-02281-f005:**
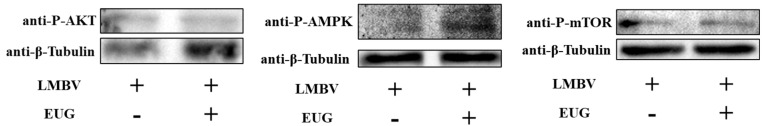
Cells were pretreated with EUG for 2 h and then infected with LMBV; cell samples were collected after 24 h and phosphorylation levels of AKT, AMPK, and mTOR proteins were determined by western blotting, with β-tubulin serving as the loading control for normalization. Data expressed as mean ± SD of three independent experiments; each experiment performed in triplicate. + indicates the addition of the virus/drug shown on the left, - indicates its absence.

**Figure 6 microorganisms-13-02281-f006:**
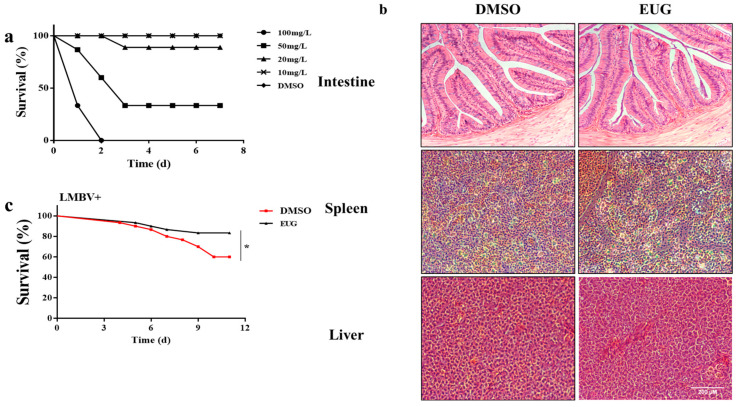
EUG reduces mortality in largemouth bass infected with LMBV. (**a**) Survival of largemouth bass after intraperitoneal injection of different concentrations of EUG (n = 45). (**b**) HE staining of healthy largemouth bass tissues treated with EUG; (**c**) Survival rate of largemouth bass after EUG injection. Data expressed as mean ± SD of three independent experiments; each experiment performed in triplicate. After confirming normal distribution, one-way ANOVA was applied for analysis; * *p* < 0.05.

**Figure 7 microorganisms-13-02281-f007:**
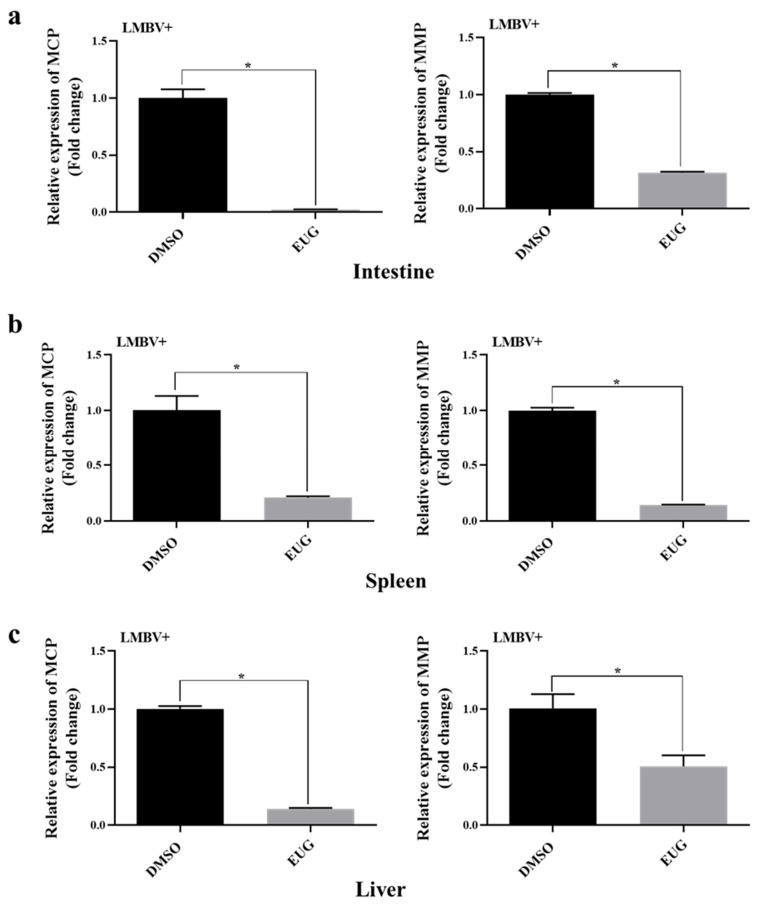
Effect of EUG on LMBV infection in largemouth bass. (**a**–**c**) Samples of various tissues of largemouth bass were collected after LMBV infection, and the mRNA expression levels of LMBV–MCP and LMBV–MMP in the intestines, spleens, and livers were detected by qRT-PCR in three individual fish. Data expressed as mean ± SD of three independent experiments; each experiment performed in triplicate. After confirming normal distribution, one-way ANOVA was applied for analysis; * *p* < 0.05.

**Figure 8 microorganisms-13-02281-f008:**
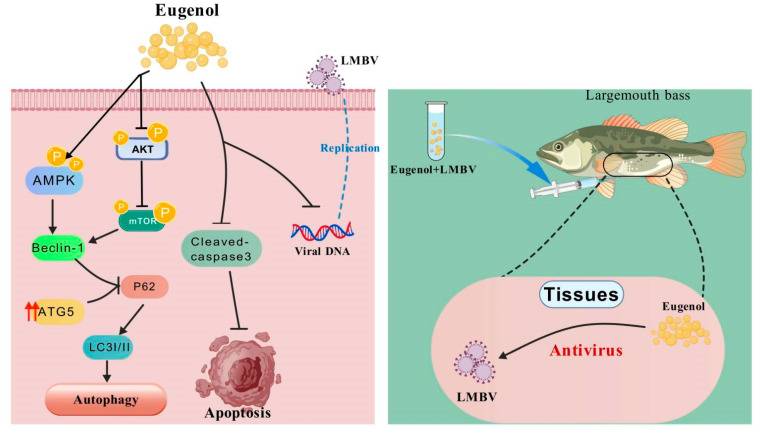
Schematic showing the antiviral activity of EUG during LMBV infection.

**Table 1 microorganisms-13-02281-t001:** Primers utilized in this research.

Primer	Sequence (5′–3′)	Amplicon Size (bp)
FHM-β-Actin-RT-F	TACGAGCTGCCTGACGGACA	195
FHM-β-Actin-RT-R	GGCTGTGATCTCCTTCTGCA	
MS-β-Actin-RT-F	CCACCACAGCCGAGAGGGAA	152
MS-β-Actin-RT-R	TCATGGTGGATGGGGCCAGG	
LMBV-MCP-RT-F	CTCGCCACTTATGACAGCCTTGAC	148
LMBV-MCP-RT-R	AACCCACGGGATAATGCTCTTTGAC	
LMBV-MMP-RT-F	GCGTATTTCGCACCCTCTG	122
LMBV-MMP-RT-R	TAAGCGTCGCCCTTGTCTG	

## Data Availability

The original contributions presented in this study are included in the article. Further inquiries can be directed to the corresponding author.

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
