# Peer review of "Antiviral Activity of Eugenol Against Largemouth Bass Ranavirus Through Regulation of Autophagy and Apoptosis In Vitro and In Vivo"

_microorganisms, 2025, doi:10.3390/microorganisms13102281_

Round 1

Reviewer 1 Report

Comments and Suggestions for Authors

Dear Authors,

Thank you for the opportunity to review your manuscript. The study addresses an important topic with potential relevance for aquaculture virology and the use of plant-derived compounds as antiviral agents. However, after a detailed evaluation of each section, I identified several academic, methodological, grammatical, and structural issues that need to be carefully addressed before the work can be considered for publication. My comments and questions are intended to provide constructive feedback to strengthen the scientific rigour, clarity, and overall quality of the manuscript.

Title

Wouldn’t a more precise title be more appropriate, for example: “Antiviral activity of Eugenol against Largemouth Bass Ranavirus through regulation of autophagy and apoptosis in vitro and in vivo”?

Abstract

The abstract is descriptive rather than critical: it lacks quantitative data (e.g., percentage reduction in viral load, statistical differences in survival).
The experimental design and the number of replicates are not specified.
Phrases such as “showing promising potential to against LMBV” are grammatically incorrect.

Introduction

Lengthy and repetitive, it accumulates general information about LMBV and eugenol without a clear delimitation of knowledge gaps.
Some sentences are poorly written, e.g., “the most common clinical symptoms… are include swelling”.
The relationship between apoptosis, autophagy, and the action of EUG is described in a scattered manner.
What is the explicit hypothesis of the research?

Materials and Methods

2.2. Fish, cells and virus

It is stated that 200 fish were used, but it is not clarified how they were distributed among treatments nor the sample size per replicate.
The recirculating water system is not described in detail (physicochemical parameters, stocking density, water renewal).
No certification of the genetic or sanitary origin of the fish is reported.
How was feeding standardised before and during the trial?
Was the viral strain genetically characterised prior to the experiment, or was only a previously isolated reference used?

2.3. Cell treatment

It is not clarified whether the stability of EUG in the culture medium during the trial was evaluated (possible degradation).
The full range of concentrations tested is not detailed, it only states “different concentrations”.
It is not specified whether the DMSO control was adjusted to the same percentage as in the EUG treatments.
Was the toxicity of the solvent (DMSO) controlled with an additional group?
Were preliminary tests conducted to determine the maximum non-cytotoxic concentration of EUG?

2.4. Cytotoxicity assays

The number of biological and technical replicates is not clearly distinguished (“eight replicates” → eight wells in a single assay, or independent experiments?).
No positive control for cytotoxicity is mentioned.
There is use of abbreviations not previously defined.
How was the 50% inhibitory concentration (IC50) of EUG calculated?
Were the results expressed as % viability relative to the control?

2.5. Virus infection assay in vitro

The multiplicity of infection (MOI), an essential parameter in viral assays, is not defined.
The rationale for choosing 2 h of pretreatment is not clear.
What criterion was used to select the time points analysed?
Was cell viability post-infection measured in addition to viral load?

2.6. RNA isolation and qRT-PCR analysis

It is not specified whether RNA purity and integrity were verified (A260/A280, RIN).
It is not mentioned whether negative controls without reverse transcriptase or without template were used.
The statement “each experimental condition was as previously described [15]” is unclear; it would be better to provide the full details.
Was expression normalised to a single reference gene (β-actin), or was the stability of more than one constitutive gene validated?

2.7. Western blotting analysis

Were positive controls included for each target protein?

2.8. Annexin V-FITC/PI Apoptosis Assay

Neither the software used nor the type of statistical analysis of early/late apoptosis is described.
Was a positive control of induced apoptosis (e.g., staurosporine) included as a reference?

2.9. Antiviral activity of EUG in vivo

How was it ensured that the reduction in mortality was due to EUG and not to individual variability differences?
Why was only the intraperitoneal route used, and not oral administration, which would be more applicable at the production level?

Results

Figures and descriptions are presented without concrete values (only “significant” without indicating exact p-values or effect size).
Why were only 50 μM and 75 μM tested in cells, if toxicity was defined up to 100 μM?
How was the dose of 10 mg/L in fish defined, if toxicity was tested up to 100 mg/L?

Discussion

Excessively long, with an overabundance of general literature review on apoptosis and autophagy, losing focus on the authors’ own findings.
Comparisons are made with other viruses and cells, but the magnitude of EUG’s effect in the present study is not interpreted.
Was the possibility considered that the observed effects may be due to a non-specific oxidative stress mechanism rather than to a specific antiviral effect?

Conclusions

It is stated that EUG “could be a potential drug”, which is too conclusive for a preliminary trial.
Should the conclusions not be nuanced, indicating that these are preliminary results and that field trials are required?

The manuscript has interesting potential, but requires corrections.

Author Response

Thank you very much for taking the time to review our manuscript and for providing these insightful comments. We greatly appreciate your efforts in helping us improve the quality of our paper. Please find our detailed responses to each of your comments below. The corresponding revisions and corrections have been carefully implemented throughout the manuscript and are highlighted in the resubmitted files by blue. 

For research article

Response to Reviewer 1 Comments

1. Summary

Thank you very much for taking the time to review our manuscript and for providing these insightful comments. We greatly appreciate your efforts in helping us improve the quality of our paper. Please find our detailed responses to each of your comments below. The corresponding revisions and corrections have been carefully implemented throughout the manuscript and are highlighted in the resubmitted files by blue.

2. Questions for General Evaluation

Reviewer’s Evaluation

Response and Revisions

Does the introduction provide sufficient background and include all relevant references?

Must be improved

Thank you very much for giving us these evaluations, we have carefully revised our manuscript to improve the quality of our work. Please see detailed responses below and the resubmitted revised manuscript.

Is the research design appropriate?

Can be improved

Are the methods adequately described?

Can be improved

Are the results clearly presented?

Must be improved

Are the conclusions supported by the results?

Must be improved

Are all figures and tables clear and well-presented?

Can be improved

3. Point-by-point response to Comments and Suggestions for Authors

Thank you for the opportunity to review your manuscript. The study addresses an important topic with potential relevance for aquaculture virology and the use of plant-derived compounds as antiviral agents. However, after a detailed evaluation of each section, I identified several academic, methodological, grammatical, and structural issues that need to be carefully addressed before the work can be considered for publication. My comments and questions are intended to provide constructive feedback to strengthen the scientific rigour, clarity, and overall quality of the manuscript. The manuscript has interesting potential, but requires corrections.

Comments 1: Title:Wouldn’t a more precise title be more appropriate, for example: “Antiviral activity of Eugenol against Largemouth Bass Ranavirus through regulation of autophagy and apoptosis in vitro and in vivo”?

Response 1: We sincerely thank the reviewer for this valuable suggestion. The proposed title, "Antiviral activity of Eugenol against Largemouth Bass Ranavirus through regulation of autophagy and apoptosis in vitro and in vivo" is indeed more precise and comprehensive. It clearly summarizes the core content of this study (the antiviral activity of eugenol) and the mechanism of action (regulation of autophagy and apoptosis), significantly enhancing the informational value and scientific rigor of the title. We fully accept this suggestion and have revised the paper title accordingly(Lines 2-4 in the resubmitted manuscript).

Comments 2: Abstract:The abstract is descriptive rather than critical: it lacks quantitative data (e.g., percentage reduction in viral load, statistical differences in survival).

The experimental design and the number of replicates are not specified.

Phrases such as “showing promising potential to against LMBV” are grammatically incorrect.

Response 2: We sincerely appreciate the reviewer's valuable feedback regarding the abstract section. Accordingly, we have revised the abstract of the manuscript to include this information (Lines 15-17 in the resubmitted manuscript). Regarding the number of experimental replicates, this information is already provided in section 2.10 of the resubmitted manuscript. Additionally, we thank the reviewer for the careful reading and have corrected the grammatical issue pointed out (Lines 12-13 in the resubmitted manuscript).

Lines 15-17: ……Firstly, EUG has been demonstrated to down-regulate both the mRNA and protein levels of the Major Capsid Protein (MCP) in LMBV-infected cells. In addition, EUG could inhibit the expression of cleaved-caspase-3 in LMBV-infected fathead minnow (FHM) cell. While EUG would not only directly regulate the protein kinase B (AKT)/mammalian target of rapamycin (mTOR) pathway, but also affect the AMP-activated protein kinase (AMPK) pathway in FHM cells during LMBV infection. Here, we demonstrate that EUG exerts its antiviral effects by modulating both LMBV-induced apoptosis and autophagy. Notably, EUG reduced the viral load present within the tissues of LMBV-infected largemouth bass, thereby ultimately enhancing their survival rate in the culture environment by about 20%.”

Lines 12-13: ……Eugenol (EUG) has potent antiviral activity, showing promising potential against LMBV.”

Comments 3: Introduction: Lengthy and repetitive, it accumulates general information about LMBV and eugenol without a clear delimitation of knowledge gaps.

Some sentences are poorly written, e.g., “the most common clinical symptoms… are include swelling”.

The relationship between apoptosis, autophagy, and the action of EUG is described in a scattered manner.

What is the explicit hypothesis of the research?

Response 3: Thank you for your suggestions. In the introduction of our manuscript, we first provide a brief overview of LMBV and the threats it poses, highlighting the urgent need to identify effective antiviral agents. We then introduce EUG and its role in modulating autophagy and apoptosis, which may influence viral replication. This is followed by a description of the relationship between viral replication and the processes of autophagy and apoptosis.

As explicitly stated in lines 85-86 of the resubmitted manuscript, we have clarified the interactions among EUG, autophagy, apoptosis, and LMBV infection. The central hypothesis of our study is to investigate whether EUG can exert anti-LMBV effects by regulating autophagy and apoptosis. Regarding the inaccuracies in sentence expression, we have carefully reviewed the manuscript and made corresponding revisions (Lines 38-40 in the resubmitted manuscrip)

Lines 85-86: ……In this study, we investigated the effectiveness of EUG against LMBV and elucidated the its possible cellular pathways such as autophagy and apoptosis.”

Lines 38-40: ……In China, the most common clinical symptoms of LMBV infection in bass are characterized by swelling of the internal organs, body surface ulceration and muscle necrosis.”

Comments 4: Materials and Methods:2.2. Fish, cells and virus

It is stated that 200 fish were used, but it is not clarified how they were distributed among treatments nor the sample size per replicate.

The recirculating water system is not described in detail (physicochemical parameters, stocking density, water renewal).

No certification of the genetic or sanitary origin of the fish is reported.

How was feeding standardised before and during the trial?

Was the viral strain genetically characterised prior to the experiment, or was only a previously isolated reference used?

Response 4: We sincerely appreciate the reviewer’s comments. Ten fish were used for viral detection. Regarding the 100 fish used in this study, they were first equally divided into five groups (n=20 per group), which received EUG injections at doses of 0 mg/kg, 10 mg/kg, 20 mg/kg, 50 mg/kg and 100 mg/kg, respectively, to determine the maximum safe dosage of eugenol in the fish. For the subsequent challenge experiment, 45 fish were assigned to each of the control and treatment groups (Section 2.2 and section 2.9 in the resubmitted manuscript).

In the recirculating aquaculture system, the water quality was maintained with dissolved oxygen >5 mg/L, a pH of 7.5 ± 0.2, and ammonia nitrogen levels around 1.0 mg/L. These parameters have been supplemented in the manuscript (Lines 101-104 in the resubmitted manuscript).

Concerning the background and health status of the experimental fish, the largemouth bass used in this study were sourced from a farm in Guangdong Province. The fish underwent viral screening and were acclimatized for approximately 10 days prior to the experiment to monitor their health. This description has been provided in the manuscript (Lines 100-101and 105-107 in the resubmitted manuscript).

Throughout the pre-experimental and experimental periods, the fish were fed a commercial diet once daily. The LMBV strain was originally isolated from largemouth bass, purified, and subsequently amplified through repeated freeze-thaw cycles of infected cells, and has been stored at -80°C(Yang et al., 2022),it was also genetically characterized prior to the experiment.

Reference:

Yang J, Xu W, Wang W, Pan Z, Qin Q, Huang X, Huang Y. Largemouth Bass Virus Infection Induced Non-Apoptotic Cell Death in MsF Cells. Viruses. 2022, 14(7):1568. 

”Two hundred largemouth bass (8.0±2 cm) were purchased from a fish farm in Guangdong Province and prepared for antiviral and toxicity analyses. The fish were cultured in a recirculating water system at 25-30 °C and fed for a 10-day acclimation period before being used in the experiments. The following water quality parameters were maintained: dissolved oxygen > 5 mg/L, pH 7.5 ± 0.2, and ammonia nitrogen at approximately 1.0 mg/L. Prior to the formal experiment, 10 fish were randomly selected for PCR detection to ensure that they were LMBV negative. LMBV was isolated from diseased largemouth bass and stored in our laboratory [11].”

Comments 5: 2.3. Cell treatment

It is not clarified whether the stability of EUG in the culture medium during the trial was evaluated (possible degradation).

The full range of concentrations tested is not detailed, it only states “different concentrations”.

It is not specified whether the DMSO control was adjusted to the same percentage as in the EUG treatments.

Was the toxicity of the solvent (DMSO) controlled with an additional group?

Were preliminary tests conducted to determine the maximum non-cytotoxic concentration of EUG?

Response 5: We sincerely appreciate the reviewer’s constructive comments. In previous experiments, we pre-incubated EUG with LMBV for 2 hours before infecting cells. The results showed no significant effect on viral titer, indicating that EUG does not directly inactivate LMBV. Furthermore, prior to each experiment, EUG was freshly prepared at appropriate concentrations according to experimental requirements to minimize inaccuracies caused by potential evaporation.

Regarding the safe concentration range of EUG, we have clearly listed the specific concentration gradients tested (eg: 50, 75, 100, 150 μM). Preliminary results indicated that the maximum non-cytotoxic concentration of EUG is below 100 μM (Section 2.4 in the resubmitted manuscript). In the “Cytotoxicity Assay” section, we have also added the following statement: “The control group was treated with an equal volume of DMSO without the addition of EUG, and the final volume concentration of DMSO in the culture medium was adjusted to 0.1% uniformly (Lines 117-119 in the resubmitted manuscript).” We initially employed the CCK-8 assay to evaluate the effects of both untreated control and 0.1% DMSO-treated groups on cell viability. The results demonstrated that compared to the untreated group, 0.1% DMSO treatment did not exert any significant impact on cellular activity. Therefore, in subsequent experiments, we utilized 0.1% DMSO as the solvent control instead of including an additional blank control group. We will ensure more rigorous experimental design in future studies. Additionally, we will further investigate whether EUG maintains its antiviral effects over 48 hours and longer exposure periods.

In section 2.4 of the resubmitted manuscript, we systematically evaluated the effects of different concentrations of eugenol (EUG) on the viability of FHM cells. This experiment was designed to determine the maximum non-cytotoxic concentration of EUG, which is essential for ensuring that subsequent antiviral assays are conducted under biologically relevant and non-detrimental conditions.

Lines 117-119: ……The control group was treated with an equal volume of DMSO without the addition of EUG, and the final volume concentration of DMSO in the culture medium was adjusted to 0.1% uniformly.”

Section 2.4: “The cytotoxic effects of EUG on FHM cells were evaluated using the Cell Counting Kit-8 (US Everbright @Inc., Suzhou, China) assay in 96-well plates. Five groups of EUG concentrations were tested: 0 μM (DMSO), 50 μM, 75 μM, 100 μM and 150 μM, with eight replicates in each set. Cells were seeded in 96-well plates at a density of 1.0×10⁴ cells per well and incubated at 28°C for 24 hours. When cell confluence reached 80–90%,cells were treated with EUG or DMSO for 24 h. Cell viability was assessed with a microplate reader (Thermo Fisher, USA) at 450 nm after 2 hours incubation in serum-free medium with Cell Counting Kit-8 and protection from light.”

Comments 6: 2.4. Cytotoxicity assays

The number of biological and technical replicates is not clearly distinguished (“eight replicates” eight wells in a single assay, or independent experiments?).

No positive control for cytotoxicity is mentioned.

There is use of abbreviations not previously defined.

How was the 50% inhibitory concentration (IC50) of EUG calculated?

Were the results expressed as % viability relative to the control?

Response 6: We sincerely thank you for reviewer’s professional comments on our manuscript. The "eight replicates" mentioned in section 2.4 refer to the eight wells within a 96-well plate, rather than eight independent experimental replicates.

In each cytotoxicity assay, both positive and negative controls were included. The abbreviations have been revised to their full terms (Lines 127-128 in the resubmitted manuscript). All experimental results were normalized by comparison with these controls. The half-maximal inhibitory concentration (IC₅₀) of EUG was calculated using the formula: IC₅₀ = 10^LogIC₅₀. The results are expressed as the percentage of cell viability relative to the control group.

Lines 127-128: ……Cell viability was assessed with a microplate reader (Thermo Fisher, USA) at 450 nm after 2 hours incubation in serum-free medium with Cell Counting Kit-8 and protection from light.”

Comments 7: 2.5. Virus infection assay in vitro

The multiplicity of infection (MOI), an essential parameter in viral assays, is not defined.

The rationale for choosing 2 h of pretreatment is not clear.

What criterion was used to select the time points analysed?

Was cell viability post-infection measured in addition to viral load?

Response 7: We sincerely appreciate the reviewers’ valuable comments. Based on our preliminary experiments, an MOI of 1 was found to induce significant cytopathic effects at 24 hours post-infection while still maintaining sufficient cell viability for evaluating the protective effects of the drug. Therefore, this MOI was selected for subsequent experiments, and relevant details have now been added to the manuscript (Lines 116-117 in the resubmitted manuscript).

The decision to pre-treat cells with EUG for 2 hours prior to viral infection was based on in vitro pre-experimental results. These data indicated that pre-incubating EUG with LMBV for 2 hours before infecting the cells did not significantly affect viral titers, suggesting that EUG does not have a direct virucidal effect against LMBV. Although these data were not presented, they informed our experimental setup.

At an MOI of 1, viral load peaks at 24 hours post-infection. Moreover, EUG is chemically unstable and prone to volatilization over extended periods (Jaganathan et al., 2012). Additionally, previous studies have shown that EUG affects cell viability and metabolism primarily within the first 24 hours (Absalan et al., 2016). Furthermore, previous research by our team has also shown that EUG exhibits anti-SGIV effects at 24 hours (Wang et al., 2024).Therefore, the 24-hour time point was selected to optimally assess the protective effects of EUG. In future studies, we systematically extend the treatment duration to 48 hours and beyond to further investigate the persistence and time-dependent characteristics of EUG’s antiviral activity.

References:

1. Jaganathan S K, Supriyanto E. Antiproliferative and molecular mechanism of eugenol-induced apoptosis in cancer cells[J]. Molecules. 2012, 17(6):6290-6304.

2. Absalan A, Mesbah-Namin SA, Tiraihi T, Taheri T. The effects of cinnamaldehyde and eugenol on human adipose-derived mesenchymal stem cells viability, growth and differentiation: a cheminformatics and in vitro study. Avicenna J Phytomed. 2016 , 6(6):643-657.

3. Wang Y, Jiang Y, Chen J, Gong H, Qin Q, Wei S. In vitro antiviral activity of eugenol on Singapore grouper iridovirus. Fish Shellfish Immunol. 2024, 151:109748.

Lines 116-117: ”……followed by LMBV infection at a multiplicity of infection (MOI) of 1 (MOI=1).”

Comments 8: 2.6. RNA isolation and qRT-PCR analysis

It is not specified whether RNA purity and integrity were verified (A260/A280, RIN).

It is not mentioned whether negative controls without reverse transcriptase or without template were used.

The statement “each experimental condition was as previously described [15]” is unclear; it would be better to provide the full details.

Was expression normalised to a single reference gene (β-actin), or was the stability of more than one constitutive gene validated?

Response 8: We sincerely thank the reviewer for raising this important question. Regarding the purity and integrity of the RNA, after extracting total RNA from cells or tissues, we assessed the samples using a NanoDrop™ One spectrophotometer. All samples exhibited A260/A280 ratios between 1.8 and 2.1. Furthermore, RNA integrity was evaluated via agarose gel electrophoresis (28S and 18S rRNA bands), and the results confirmed that all RNA samples were intact and suitable for subsequent experiments.

To ensure the reliability of the qRT-PCR results, we included stringent negative controls in each experimental batch: a no-reverse transcription control (-RT) to rule out genomic cDNA contamination, and a no-template control (NTC) to monitor potential reagent contamination.

In our practice, we routinely use a dual endogenous reference gene system comprising 18S rRNA and β-actin for normalization to enhance the accuracy and reproducibility of quantitative results. This strategy helps minimize biases that may arise from fluctuations in the expression of a single housekeeping gene or experimental variations. In the manuscript, We only displayed the results normalized to β-actin, as it demonstrated consistent and reliable expression under our experimental conditions. Moreover, we have supplemented detailed experimental conditions in section 2.6 of the resubmitted manuscript.

“Tissues were ground in pre-cooled PBS with a homogenizer. Total RNA was isolated using the Animal/Cell Total RNA Isolation Kit (FOREGENE, Chengdu, China) and reverse-transcribed with the ReverTra Ace qPCR RT Kit (TOYOBO) [22]. At the indicated time points, LMBV-infected cells were harvested for RNA extraction as described previously[15]. In brief, the total RNA of infected cells was extracted using a cell total RNA isolation kit according to the manufacturer’s protocol. The RNA was reverse transcribed using a ReverTra Ace a kit (Toyobo, Osaka, Japan). The transcription level of viral genes, including MCP and myristoylated membrane protein (MMP), were examined using qRT-PCR analysis. qRT-PCR was performed using SYBR® Green Realtime PCR Master Mix (Toyobo, Osaka, Japan) in an Applied Biosystems QuantStudio 5 Real Time detection system (Thermo Fisher Scientific) as described previously. The cycling conditions for qRT-PCR were as follows: 95 for 5 min for activation followed by 45 cycles at 95 for 5 s, 60 for 10 s, and 72 for 15 s. The primers utilized for the PCR are detailed in Table 1. Moreover, the primers used in this experiment had annealing temperatures ranging approximately between 58-65°C, and were validated by melt curve analysis and standard curve quantification, demonstrating amplification efficiencies of 96-100% with R² > 0.96. The 2-ΔΔCT method was used to calculate the relative expression ratio of the selected genes to β-actin (reference gene). Statistics were calculated using SPSS version 20. Differences were considered to statistically significant when *p < 0.05.

Comments 9: 2.7. Western blotting analysis

Were positive controls included for each target protein?

Response 9: Thank you very much for raising this valuable suggestion. We fully understand and appreciate the importance of including positive controls to strengthen the persuasiveness of western blotting results. However, in the current study, we decided not to include separate positive controls for each target protein for the following reasons:

First, the primary objective of this study was to compare the relative changes in protein expression among different treatment groups within the same cellular model. Our experimental design already includes a comprehensive control system (eg: DMSO control group, EUG treatment group, LMBV infection group, and EUG + LMBV co-treatment group). These groups serve as internal references that mutually validate one another, allowing clear and reliable interpretation of trends in target protein expression under various conditions.

Moreover, the expression stability of all key proteins examined in this study has been systematically validated with positive controls in numerous relevant publications (Deng et al., 2022; Yang et al., 2022). These previous studies not only confirmed the highly consistent expression of these proteins under similar experimental conditions but also ruled out potential variations caused by factors such as cell type, treatment procedures, or time points. This existing evidence strongly supports the reliability and reproducibility of the protein indicators used in our experimental system. Furthermore, all primary antibodies used in this study underwent rigorous specificity validation by the manufacturers. Their applicability was further confirmed in preliminary experiments conducted in our laboratory. The presence of single, clear bands in the Western Blot results serves as strong evidence of antibody specificity. Finally, all Western Blot results were verified through multiple independent biological replicates, which further ensures the robustness of our findings.

We sincerely appreciate your feedback, which provides important guidance for the design of our future studies. We will certainly consider incorporating positive controls in subsequent experiments as suggested.

References:

1. Deng L, Feng Y, OuYang P, Chen D, Huang X, Guo H, Deng H, Fang J, Lai W, Geng Y.  Autophagy induced by largemouth bass virus inhibits virus replication and apoptosis in Epithelioma papulosum cyprini cells. Fish Shellfish Immunol. 2022, 123, 489-495.

2. Yang J, Xu W, Wang W, Pan Z, Qin Q, Huang X, Huang Y. Largemouth Bass Virus Infection Induced Non-Apoptotic Cell Death in MsF Cells. Viruses. 2022, 14(7):1568

Comments 10: 2.8. Annexin V-FITC/PI Apoptosis Assay

Neither the software used nor the type of statistical analysis of early/late apoptosis is described.

Was a positive control of induced apoptosis (e.g., staurosporine) included as a reference?

Response 10: We thank the reviewer for these important and professional comments. We have supplemented the apoptosis analysis methodology in the manuscript. Specifically, the statistical analysis of apoptosis rates was performed using FlowJo™ software (v10.8.1). Annexin V-FITC/PI dual staining flow cytometry was employed to distinguish cell populations by gating on scatter plots: viable cells (Annexin V⁻/PI⁻), early apoptotic cells (Annexin V⁺/PI⁻), and late apoptotic/necrotic cells (Annexin V⁺/PI⁺). The percentages of each cell population were quantified for statistical analysis. These data have been incorporated into section 2.8 of the resubmitted manuscript.

For the positive control of induced apoptosis, Firstly, previously published studies from our laboratory have confirmed that LMBV effectively induces apoptosis in FHM cells, thereby providing a well-established positive control model for this experiment (Yang et al.,2022). Based on the experimental framework established with chemical positive controls, this study specifically investigated how EUG modulates LMBV-induced apoptosis in FHM cells by analyzing relative apoptotic changes across treatment groups. Our design incorporated a comprehensive control system, including: DMSO treatment, eugenol (EUG) treatment, LMBV infection treatment, and EUG + LMBV co-treatment. Comparative analysis revealed that LMBV infection effectively induced apoptosis, with significantly elevated apoptosis rates compared to baseline levels. To ensure robust statistical power, we prioritized sufficient biological replication between viral infection and drug intervention groups. All findings were consistently verified through multiple independent experimental replicates.

Reference:

Yang J, Xu W, Wang W, Pan Z, Qin Q, Huang X, Huang Y. Largemouth Bass Virus Infection Induced Non-Apoptotic Cell Death in MsF Cells. Viruses. 2022; 14(7):1568.

“……Each sample was measured in triplicate. The obtained data were analyzed using Flow-Jo 10.4 software, and cell populations were differentiated via Annexin V-FITC/PI dual-parameter gating into three distinct groups: viable (Annexin V⁻/PI⁻), early apoptotic (Annexin V⁺/PI⁻), and late apoptotic/necrotic (Annexin V⁺/PI⁺) cells, with proportional data extracted for statistical comparisons. The results presented were from one representative experiment.”

Comments 11: 2.9. Antiviral activity of EUG in vivo

How was it ensured that the reduction in mortality was due to EUG and not to individual variability differences?

Why was only the intraperitoneal route used, and not oral administration, which would be more applicable at the production level?

Response 11: We fully appreciate the reviewer’s valuable comments regarding individual variability and the route of drug administration. In the present study, several methodological features were implemented to ensure that the observed reduction in mortality could be robustly attributed to EUG treatment rather than to individual differences. These included a completely randomized experimental design, an adequate sample size (n= 45 per group), the inclusion of stringent parallel control groups, and a clear dose-dependent effect. Together, these measures significantly reduced the impact of chance and enhanced the attributability and reliability of the results.

At this foundational research stage, intraperitoneal injection was selected as the route of administration based on support from existing literature (Jeong et al., 2015). This approach was chosen primarily to ensure sufficient drug absorption and bioavailability, thereby allowing us to directly address the core scientific question: “Does EUG exert protective effects in vivo?” By avoiding uncertainties associated with oral absorption and metabolism, we reduced the risk of confounding factors that could obscure interpretation. Such as the inability to distinguish between true drug ineffectiveness and delivery failure. Our results demonstrate a clear protective effect of EUG in vivo, which provides a solid foundation for future development of more practical administration regimens including oral formulations.

Reference:

Jeong KH, Lee DS, Kim SR. Effects of eugenol on granule cell dispersion in a mouse model of temporal lobe epilepsy. Epilepsy Res. 2015, 115:73-6.

Comments 12: Results: Figures and descriptions are presented without concrete values (only “significant” without indicating exact p-values or effect size).

Why were only 50 μM and 75 μM tested in cells, if toxicity was defined up to 100 μM?

How was the dose of 10 mg/L in fish defined, if toxicity was tested up to 100 mg/L?

Response 12: We thank the reviewer for their valuable comments. In our manuscript, all figure and table legends have been updated to include specific statistical p-values (denoted as *p < 0.05).

The selection of EUG concentrations was based on previous literature and preliminary experiments. Cell viability was measured using the CCK-8 assay after 24 hours of treatment with various concentrations of eugenol (0, 25, 50, 75, 100, and 150 μM) to determine the half-maximal inhibitory concentration (IC₅₀). The results showed that at 100 μM, cell viability decreased to approximately 90% compared to the control group, indicating that this concentration represents a threshold for significant cytotoxicity. Therefore, all subsequent antiviral experiments were conducted using subtoxic concentrations below the IC₅₀ to exclude interference from cytotoxic effects.

To evaluate the maximum safe concentration of EUG in fish, we administered different injection doses (0, 10, 20, 50, and 100 mg/kg) to largemouth bass. The results demonstrated that fish injected with 10 mg/kg EUG remained healthy with no mortality for up to seven days. In contrast, a dose of 100 mg/kg resulted in 100% mortality within two days. Accordingly, to investigate the anti-LMBV effects of EUG in vivo without confounding toxicity, we selected the non-toxic dose of 10 mg/kg EUG for all subsequent injection experiments.

Comments 13: Discussion: Excessively long, with an overabundance of general literature review on apoptosis and autophagy, losing focus on the authors’ own findings.

Comparisons are made with other viruses and cells, but the magnitude of EUG’s effect in the present study is not interpreted.

Was the possibility considered that the observed effects may be due to a non-specific oxidative stress mechanism rather than to a specific antiviral effect?

Response 13: We sincerely thank you for your valuable feedback. In the resubmitted manuscript, we have added relevant content to the discussion section. In response, we have revised the manuscript to provide a more focused interpretation. Specifically, at lines 489 of the resubmitted manuscript, we have added data clarifying the in vivo efficacy of EUG.

To further address the question of specific antiviral mechanisms, we would like to clarify two key points based on our experimental work: In previous in vitro experiments, pre-incubation of LMBV with EUG for 2 hours prior to cell infection showed no significant effect on viral titer, indicating that EUG does not directly inactivate LMBV particles. Furthermore, during our assessment of the safe concentration of EUG in largemouth bass (Micropterus salmoides), we included a positive oxidative stress control group. Importantly, compared to this control, fish injected with EUG exhibited normal health status without any apparent damage or mortality. This observation suggests that the observed antiviral effects are more likely attributable to a specific mechanism rather than non-specific oxidative stress.

“……The survival rate of fish in EUG treatment group increased by about 20% compared to the control group,……”

Comments 14: Conclusions: It is stated that EUG “could be a potential drug”, which is too conclusive for a preliminary trial.

Should the conclusions not be nuanced, indicating that these are preliminary results and that field trials are required?

Response 14: We sincerely appreciate reviewer’s suggestion. We agree that based on the preliminary findings from our cell-based model, directly describing EUG as "a potential therapeutic agent" was overly optimistic and definitive. In accordance with your feedback, we have revised the relevant concluding statements throughout the manuscript (specifically at lines 511-513 in the resubmitted manuscript) to adopt a more cautious and measured tone.

……This work positions EUG as a promising therapeutic candidate and provides a novel theoretical framework for its future application in antiviral strategies.”

4. Response to Comments on the Quality of English Language

Point 1: The English could be improved to more clearly express the research.

Response: Thank you very much for giving us this suggestion, we have carefully revised our manuscript to improve the quality of language. Please see the resubmitted revised manuscript with the revision sections and content highlighted by blue.

Reviewer 2 Report

Comments and Suggestions for Authors

This study provides new insight into the antiviral potential of eugenol against largemouth bass ranavirus, an area with limited prior investigation. The methodological breadth enhances the reliability of the conclusions and provides multiple independent lines of evidence. The integration of molecular markers with survival analysis demonstrates a thoughtful design. Please try to incorporate the comments to improve the manuscript

Major concern

  1. Statistical design and reporting are insufficient for in vivo work: No a priori power analysis, no randomization/blinding details, unclear experimental unit, survival analysis missing hazard/risk estimates; multiple t-tests and one-way ANOVA are referenced without correction for multiple comparisons. Define the experimental unit (fish vs tank), randomization/blinding, and perform power calculations. For survival, use Kaplan–Meier curves with log-rank test plus Cox proportional hazards (hazard ratio with 95% CI). Adjust p-values across tissues/genes (Benjamini–Hochberg). Report exact n, effect sizes, and CIs for all tests.
  2. Controls are incomplete for mechanistic claims about AMPK and AKT/mTOR: The conclusion that EUG “regulates AMPK and AKT/mTOR” rests only on phospho-western blots without pathway perturbation (no AMPK activator/inhibitor, AKT/mTOR inhibitors, or siRNA/CRISPR validation).  Add pharmacological controls (such as, Compound C for AMPK inhibition, AICAR for activation; MK-2206 for AKT; rapamycin/torin for mTOR) and/or genetic knockdown to test whether EUG’s antiviral effect is abrogated when these pathways are blocked.
  3. qRT-PCR normalization and primer reporting are incomplete: Single reference gene (β-actin) is used across different tissues and treatments without stability validation; primer table is truncated (“Error! Reference source not found.”) and lacks amplicon sizes/efficiencies (L120–129). Validate ≥2 reference genes (for example, ef1α, rpl13a, gapdh) with geNorm/NormFinder per tissue/condition; include amplification efficiency (standard curves), melt curves, and amplicon lengths. Correct/restore the full primer table.
  4. In vivo dosing and exposure route lack pharmacology/toxicity context: A single intraperitoneal dose (10 mg/L in 100 µL) is used; no dose–response, PK/PD rationale, or tissue distribution; “safe concentration” is based only on 7-day mortality and histology (L158–171, L266–280). Provide mg/kg dosing, justify with literature, test multiple doses (low/med/high), and include sublethal endpoints (behavior, cortisol, hematology). If feasible, add simple PK (plasma/time) or at least time-to-peak effect; extend safety observation beyond 7 days.
  5. Writing/format issues reduce clarity and reproducibility: Numerous grammatical errors; duplicated “Institutional Review Board Statement” sections with conflicting content (L419–427 vs L426–427); missing figure legends’ details (replicates, statistics); some references are outdated/mis-cited (e.g., “Originally from the Mississippi River system in California,” L294 — geographically confusing); and a persistent formatting error in the Methods (“Error! Reference source not found.”) (L294–299, L419–427). Thorough language edit; remove duplicate statements and ensure ethics text matches Methods (L88–91); correct geographic phrasing; repair cross-references; expand figure legends with n, stats, and exact tests.

Minor concern

L10-12, “severely harmful… high mortality rate” → tighten: “causes high mortality during outbreaks.”

L13-19, Claim sequence is unclear; specify the hypothesis and primary outcomes. Replace “to investigated” with “to investigate.”

L15-18, Mechanism claims (AKT/mTOR↓, AMPK↑) should be toned down to “associated with” unless you include inhibitor/activator data.

L20-23, Provide quantitative effect sizes: e.g., survival +20 percentage points (60%→80%), fold-change reductions in MCP/MMP with ±SD. Cite in vivo n.

L23-24, “significantly enrich the content of antiviral research” — vague; replace with specific aquaculture relevance.

L27-34, Clarify: LMBV is an iridovirus (genus Ranavirus), first described in the US; remove “Mississippi River system in California” (later) to avoid confusion; ensure temperature range phrasing is “25–30 °C.” 

L42-51, Nicely frames apoptosis/autophagy; add a crisp concluding paragraph stating your study aims and hypotheses.

L52-62, Good background on EUG; add a balanced note on EUG’s known toxicity limits and solvent considerations (DMSO vehicle).

L63–78, Distinguish cancer-cell autophagy literature from antiviral contexts; avoid over-gener-lization.

L79-86, End with a sharper “Here we show…” sentence listing models (FHM, largemouth bass), readouts (MCP/MMP qPCR; westerns; survival).

L87-91, Ensure species and anesthesia/euthanasia methods are stated.

L92-101, Report fish mass/length (mean±SD), tank numbers, stocking density, photoperiod, water quality. Define acclimation duration precisely (“10 days”). State LMBV strain, passage number, and titer method. 

L102-108, Express EUG concentration in μM for cells; specify final DMSO % (keep ≤0.1–0.2%) and include a vehicle control at the same %.

L109-115, Include exact seeding density, incubation time before treatment, background subtraction method, and normalization. Add IC50 if relevant.

L116-119, State MOI, adsorption time, and wash steps.

Table 1 (L120-121), Provide a complete table: gene, primer sequences, amplicon size, efficiency (%), R², annealing temp. 

qRT-PCR (L121-129), Use MIQE-compliant reporting: reference gene validation, efficiency correction (Pfaffl), melt curves, no-RT/no-template controls.

Western blot (L130-148), Add protein load per lane (μg), membrane pore size, blocking buffer composition, antibody catalog numbers, and exposure/quantification method (e.g., ImageJ), normalization strategy (β-tubulin validation).

Annexin-V/PI (L149-157), Specify cell number per tube, gating strategy, instrument model/settings, and positive control (staurosporine).

In vivo antiviral (L158-171), Convert “mg/L in 100 μL” to mg/kg dose; describe randomization, blinding of outcome assessment, infection dose rationale (LD50 or TCID50 per fish), housing/tank as unit, and humane endpoints.

Statistics (L172-176), Pre-specify primary endpoints; use KM/log-rank for survival, mixed models for repeated qPCR if applicable; report effect sizes and FDR-adjusted p-values.

L178-190 (3.1), Report n (wells/experiments) and effect sizes for CCK-8. For qPCR, give fold-change ±SD and exact p-values; add MOI.

Fig.1 legend (L192-196), Include n, stats (test, two-sided), and whether normality was assessed.

L197-204 (3.2), Provide time-course stats (two-way ANOVA with interaction: treatment×time). Add representative blots with full lane order and molecular weights.

Fig.2 legend (L206-211), State biological replicates (not just technical), normalization method for westerns, and loading control validation.

L212-225 (3.3), Include quantification of cleaved-caspase-3 normalized to β-tubulin with densitometry (mean±SD, n, stats). Add representative dot plots for flow cytometry and define early vs late apoptosis gates.

Fig.3 legend (L233-237), Add positive control (e.g., staurosporine) and gating strategy.

L238-247 (3.4): Autophagy markers: add bafilomycin A1 or chloroquine flux assay to demonstrate increased autophagic flux (not just accumulation).

Fig.4 legend (L249-251), State whether flux inhibitors were used; otherwise rephrase claims to “autophagy marker profile consistent with increased flux.”

L252-261 (3.5), Pathway causality is not established; add inhibitor/activator experiments.

Fig.5 legend (L262-265), Report normalized densitometry (p-AKT/AKT, p-mTOR/mTOR, p-AMPK/AMPK).

L266-276 (3.6), Survival: present KM curves with log-rank p and HR (95% CI). Provide per-group n, censoring, and humane endpoints. Clarify whether 12-day endpoint was pre-specified.

Fig.6 legend (L278-280), Add scale bars for histology, staining details, and blinded scoring if used.

L283-291 (3.7), Tissue qPCR: show per-fish data (points), normalize with validated reference genes, and report effect sizes + FDR across tissues/targets.

Fig.7 legend (L289–291), Add n (fish), technical vs biological replicates, and correction for multiple testing.

L294-299, “Originally from the Mississippi River system in California” — geographically inconsistent; revise to “native to the Mississippi River basin; introduced widely…” with authoritative citation. 

L300-308, Good overview of Iridoviridae; add a sentence acknowledging conflicting reports on autophagy’s role in different cell types/species.

L309-326, When asserting “75 μM significantly inhibited…”, remind MOI and provide fold-change numbers; caution that cell culture solvent (DMSO) controls matched.

L327-352, Nicely ties apoptosis literature; avoid implying generality beyond tested systems; specify that your evidence for apoptosis inhibition is limited to FHM at 12–24 h.

L353-377, For autophagy, explicitly note the lack of flux inhibitor experiments and present as a limitation.

L378-396, Pathway section should acknowledge that phosphorylation changes are correlative; state the need for pathway perturbation.

L397-403, Final paragraph: integrate quantitative highlights (survival HR; tissue viral load fold-changes) and a forward-looking statement on dosing studies and delivery (bath/oral, not only IP). Trim to one tight paragraph; avoid restating results; emphasize practical next steps (dose-response, route optimization, safety, and regulatory path)

Author Response

Thank you very much for taking the time to review our manuscript and for providing these insightful comments. We greatly appreciate your efforts in helping us improve the quality of our paper. Please find our detailed responses to each of your comments below. The corresponding revisions and corrections have been carefully implemented throughout the manuscript and are highlighted in the resubmitted files by blue.

The Original images of Western Blotting have been uploaded as Supplementary File. If you can not find this file, please feel free to contact editors or us.

For research article

Response to Reviewer 2 Comments

1. Summary

Thank you very much for taking the time to review our manuscript and for providing these insightful comments. We greatly appreciate your efforts in helping us improve the quality of our paper. Please find our detailed responses to each of your comments below. The corresponding revisions and corrections have been carefully implemented throughout the manuscript and are highlighted in the resubmitted files by blue.

2. Questions for General Evaluation

Reviewer’s Evaluation

Response and Revisions

Does the introduction provide sufficient background and include all relevant references?

Can be improved

Thank you very much for giving us these evaluations, we have carefully revised our manuscript to improve the quality of our work. Please see detailed responses below and the resubmitted revised manuscript.

Is the research design appropriate?

Can be improved

Are the methods adequately described?

Can be improved

Are the results clearly presented?

Can be improved

Are the conclusions supported by the results?

Can be improved

Are all figures and tables clear and well-presented?

Can be improved

3. Point-by-point response to Comments and Suggestions for Authors

This study provides new insight into the antiviral potential of eugenol against largemouth bass ranavirus, an area with limited prior investigation. The methodological breadth enhances the reliability of the conclusions and provides multiple independent lines of evidence. The integration of molecular markers with survival analysis demonstrates a thoughtful design. Please try to incorporate the comments to improve the manuscript

Major concern

Comments 1: Statistical design and reporting are insufficient for in vivo work: No a priori power analysis, no randomization/blinding details, unclear experimental unit, survival analysis missing hazard/risk estimates; multiple t-tests and one-way ANOVA are referenced without correction for multiple comparisons. Define the experimental unit (fish vs tank), randomization/blinding, and perform power calculations. For survival, use Kaplan–Meier curves with log-rank test plus Cox proportional hazards (hazard ratio with 95% CI). Adjust p-values across tissues/genes (Benjamini–Hochberg). Report exact n, effect sizes, and CIs for all tests.

Response 1: We sincerely thank the reviewer for their thorough and professional assessment. In fact, we conducted a power analysis to determine the final sample size for the experiment. Detailed justification regarding the sample size for the in vivo experiment has been provided in section 2.9 of the resubmitted manuscript. A total of 200 fish were used in one independent in vivo study in this research. Among them, 10 fish were allocated for virus detection prior to the formal experiment, 100 fish were used to determine the safe concentration of EUG in largemouth bass (20 fish per group), and the remaining 90 fish were utilized to verify whether EUG exhibits antiviral effects in largemouth bass. All the aforementioned experiments were independently repeated three times. The experimental unit in this study was individual fish, with each subject independently treated and measured for all indicators. While random allocation was implemented during group assignment in the in vivo experiments.

To evaluate the effect of EUG on the survival rate of largemouth bass following LMBV infection, we performed Kaplan–Meier survival analysis. The log-rank test indicated a statistically significant difference (p < 0.05) in survival curves between the DMSO + LMBV treatment group and the EUG + LMBV treatment group at twelve days post-infection. In future studies, we will expand the sample size to include largemouth bass at different developmental stages, as well as wild and farmed populations, and employ the Cox proportional hazards model to further assess the in vivo anti-LMBV efficacy of EUG. All experimental data underwent multiple comparisons using two/one-way ANOVA, as appropriate for each analysis.

Comments 2: Controls are incomplete for mechanistic claims about AMPK and AKT/mTOR: The conclusion that EUG “regulates AMPK and AKT/mTOR” rests only on phospho-western blots without pathway perturbation (no AMPK activator/inhibitor, AKT/mTOR inhibitors, or siRNA/CRISPR validation). Add pharmacological controls (such as, Compound C for AMPK inhibition, AICAR for activation; MK-2206 for AKT; rapamycin/torin for mTOR) and/or genetic knockdown to test whether EUG’s antiviral effect is abrogated when these pathways are blocked.

Response 2: We sincerely thank the reviewer for their thorough assessment and highly valuable professional comments. As demonstrated in previous studies (Wang et al., 2024; Wang et al., 2020), changes in the levels of key signaling molecules, including both total and phosphorylated forms of p-AMPK/AMPK, p-AKT/AKT, and p-mTOR/mTOR can provide preliminary evidence for the activation or inhibition status of the AMPK and AKT/mTOR signaling pathways.

We fully agree with the reviewer's suggestion that employing pathway-specific agonists or inhibitors would significantly help elucidate the mechanism by which EUG exerts its anti-LMBV effects. In the current study, we primarily inferred EUG's potential antiviral effects through modulation of these pathways based on changes in phosphorylation levels. In future studies, we will utilize activators or inhibitors of the AMPK and AKT/mTOR pathways (such as Compound C for AMPK inhibition, AICAR for activation; MK-2206 for AKT; rapamycin or torin for mTOR) and genetic knockdown approaches to further investigate the anti-LMBV mechanisms of EUG. These experiments will contribute to a more comprehensive understanding of how EUG exerts its antiviral effects against LMBV by regurating AMPK and AKT/mTOR pathways .

References:

1. Wang W, Zheng Z, Qi X, Wei H, Mao X, Su Q, Chen X, Feng Y, Qiao G, Ma T, Tang Z, Zhou G, Zhuang J, Zhang P. Clinical efficacy of Fufang Yinhua Jiedu (FFYH) granules in mild COVID-19 and its anti-SARS-CoV-2 mechanism by blocking autophagy through inhibiting the AKT/mTOR signaling pathway. Front Pharmacol. 2024, 16, 15:1431617.

2. Wang X, Lin Y, Kemper T, Chen J, Yuan Z, Liu S, Zhu Y, Broering R, Lu M. AMPK and AKT/mTOR signalling pathways participate in glucose-mediated regulation of hepatitis B virus replication and cellular autophagy. Cell Microbiol. 2020, 22(2):e13131.

Comments 3: qRT-PCR normalization and primer reporting are incomplete: Single reference gene (β-actin) is used across different tissues and treatments without stability validation; primer table is truncated (“Error! Reference source not found.”) and lacks amplicon sizes/efficiencies (L120–129). Validate ≥2 reference genes (for example, ef1α, rpl13a, gapdh) with geNorm/NormFinder per tissue/condition; include amplification efficiency (standard curves), melt curves, and amplicon lengths. Correct/restore the full primer table.

Response 3: We sincerely appreciate the time and effort you have dedicated to reviewing our manuscript, as well as your valuable and professional comments. Regarding the incomplete primer table, this was due to an oversight during document editing, and we have now corrected it (Line 150 in the resubmitted manuscript).

We fully understand the importance of validating the stability of reference genes under different experimental conditions. In our study, to ensure the accuracy and reliability of quantitative results, we typically employ a dual-reference gene system consisting of 18S rRNA and β-actin for normalization. In the resubmitted manuscript, we only included results normalized to β-actin, as it demonstrated consistent and reliable expression under our experimental conditions and served as a representative reference.

In the resubmitted manuscript, the amplification efficiency of all primers was determined using standard curves generated from 10-fold serial dilutions of template cDNA. All target genes exhibited amplification efficiencies between 96% and 100% (R² > 0.96). Furthermore, melting curve analysis confirmed single-peak for all amplification products without non-specific products or primer-dimers. The tables in our manuscript include gene names and primer sequences, with the addition of amplicon sizes, ensuring optimal qPCR performance (Table 1). This section has been added to lines 151-153 of the resubmitted manuscript:

……The primers utilized for the PCR are detailed in Table 1. Moreover, the primers used in this experiment had annealing temperatures ranging approximately between 60-65°C, and were validated by melt curve analysis and standard curve quantification, demonstrating amplification efficiencies of 96-100% with R² > 0.96.”

Comments 4: In vivo dosing and exposure route lack pharmacology/toxicity context: A single intraperitoneal dose (10 mg/L in 100 µL) is used; no dose–response, PK/PD rationale, or tissue distribution; “safe concentration” is based only on 7-day mortality and histology (L158–171, L266–280). Provide mg/kg dosing, justify with literature, test multiple doses (low/med/high), and include sublethal endpoints (behavior, cortisol, hematology). If feasible, add simple PK (plasma/time) or at least time-to-peak effect; extend safety observation beyond 7 days.

Response 4: Thank you for your pharmacological review of our manuscript. For the dose response in the in vivo experiment, we intraperitoneally injected different concentrations of EUG to determine its maximum safe concentration in largemouth bass. The results showed that at a dose of 10 mg/kg, the fish remained in good health and stable condition for one month. While the manuscript only presents survival data at 7 days post-treatment, future studies will extend the observation period to two months or longer to evaluate the effects of different EUG concentrations on the survival of largemouth bass. Furthermore, fish of varying size will be employed to further characterize the in vivo dose-response relationship, pharmacokinetic/pharmacodynamic (PK/PD), and sublethal effects of EUG.

Comments 5: Writing/format issues reduce clarity and reproducibility: Numerous grammatical errors; duplicated “Institutional Review Board Statement” sections with conflicting content (L419–427 vs L426–427); missing figure legends’ details (replicates, statistics); some references are outdated/mis-cited (e.g., “Originally from the Mississippi River system in California,” L294 — geographically confusing); and a persistent formatting error in the Methods (“Error! Reference source not found.”) (L294–299, L419–427). Thorough language edit; remove duplicate statements and ensure ethics text matches Methods (L88–91); correct geographic phrasing; repair cross-references; expand figure legends with n, stats, and exact tests.

Response 5: Thank you very much for your comments on our initial draft. We have corrected the grammatical errors identified in the text and removed redundant or conflicting sentences. Regarding the inconsistency in ethical statements within the manuscript, we have revised the relevant sections to ensure complete uniformity throughout. We have removed the “Institutional Review Board Statement: Not applicable.” (Lines 519-527 in the resubmitted manuscript).

In cases where inaccuracies were identified in the cited references, such as errors regarding geographical locations, we have carefully verified and corrected these mistakes (Lines 364-365 in the resubmitted manuscript). Detailed information pertaining to figure captions, including the number of experimental replicates and statistical methods, has been clarified in section 2.10 of the manuscript. Additionally, the sample size and specific statistical tests have been indicated in each figure legend. We have also rectified formatting errors in the cross-referencing of the Methods section.

Lines 364-365: “Largemouth bass is widely distributed throughout the freshwater waters of the United States and Canada [23]

Minor concern

Comments 6: L10-12, “severely harmful… high mortality rate” → tighten: “causes high mortality during outbreaks.”

Response 6: Yes, thank you for your suggestion. We have revised the corresponding content in lines 11–12 of the resubmitted manuscript accordingly.

……Largemouth bass ranavirus (LMBV) causes high mortality rate in largemouth bass during outbreaks, resulting in huge economic losses.”

Comments 7: L13-19, Claim sequence is unclear; specify the hypothesis and primary outcomes. Replace “to investigated” with “to investigate.”

Response 7: In the resubmitted manuscript, we have reorganized the content and adjusted the sequence. Specifically, lines 12–14 in the resubmitted manuscript now present the hypothesis of this study, while lines 14-23 in the resubmitted manuscript summarize the key results. Any grammatical errors identified in this section have also been corrected (Lines 13 in the resubmitted manuscript).

……Eugenol (EUG) has potent antiviral activity, showing promising potential against LMBV. Thus, to investigate EUG’s efficacy against LMBV, corresponding analysis was conducted in vivo and in vitro. Firstly, EUG has been demonstrated to down-regulate both the mRNA and protein levels of the Major Capsid Protein (MCP) in LMBV-infected cells. In addition, EUG could inhibit the expression of cleaved-caspase-3 in LMBV-infected fathead minnow (FHM) cell. While EUG would not only directly regulate the protein kinase B (AKT)/mammalian target of rapamycin (mTOR) pathway, but also affect the AMP-activated protein kinase (AMPK) pathway in FHM cells during LMBV infection. These results indicated EUG exerts its antiviral effects by modulating both LMBV-induced apoptosis and autophagy. Notably, EUG reduced the viral load present within the tissues of LMBV-infected largemouth bass, thereby ultimately enhancing their survival rate in the culture environment by about 20%.”

Comments 8: L15-18, Mechanism claims (AKT/mTOR↓, AMPK↑) should be toned down to “associated with” unless you include inhibitor/activator data.

Response 8: Yes, we have revised the corresponding description in the manuscript (Lines 17-20 in the resubmitted manuscript).

……While EUG would not only regulate the protein kinase B (AKT)/mammalian target of rapamycin (mTOR) pathway, but also affect the AMP-activated protein kinase (AMPK) pathway in fathead minnow (FHM) cells during LMBV infection.”

Comments 9: L20-23, Provide quantitative effect sizes: e.g., survival +20 percentage points (60%→80%), fold-change reductions in MCP/MMP with ±SD. Cite in vivo n.

Response 9: We have added supplementary content in lines 14–23 of the resubmitted manuscript. We have revised the description as follows:

“……Firstly, EUG has been demonstrated to down-regulate both the mRNA and protein levels of the Major Capsid Protein (MCP) in LMBV-infected cells. In addition, EUG could inhibit the expression of cleaved-caspase-3 in LMBV-infected fathead minnow (FHM) cell. While EUG would not only directly regulate the protein kinase B (AKT)/mammalian target of rapamycin (mTOR) pathway, but also affect the AMP-activated protein kinase (AMPK) pathway in FHM cells during LMBV infection. These results indicated EUG exerts its antiviral effects by modulating both LMBV-induced apoptosis and autophagy. Notably, EUG reduced the viral load present within the tissues of LMBV-infected largemouth bass, thereby ultimately enhancing their survival rate in the culture environment by about 20%.”

Comments 10: L23-24, “significantly enrich the content of antiviral research” — vague; replace with specific aquaculture relevance.

Response 10: In lines 25-27 of the resubmitted manuscript, we have revised the description as follows:

"……These mechanistic assays revealed the anti-LMBV properties of EUG, which could sig-nificantly enrich the research content of plant extracts in the field of aquatic antiviral, and provide important theoretical basis for the development and application of related products."

Comments 11: L27-34, Clarify: LMBV is an iridovirus (genus Ranavirus), first described in the US; remove “Mississippi River system in California” (later) to avoid confusion; ensure temperature range phrasing is “25–30 °C.”

Response 11: We also identified the inconsistency in the manuscript and have deleted the inaccurate description in the subsequent section (Lines 364-365 in the resubmitted manuscript). The temperature range has been revised to "25–30°C" throughout the text(Line 38 in the resubmitted manuscript)

Lines 364-365: Largemouth bass is widely distributed throughout the freshwater waters of the United States and Canada [23].

Line 38: ……LMBV is most prevalent in summer, with outbreaks more likely at 25-30 °C.”

Comments 12: L42-51, Nicely frames apoptosis/autophagy; add a crisp concluding paragraph stating your study aims and hypotheses.

Response 12: The aims and hypothesis of the study are stated in lines 83-86 in the resubmitted manuscript.

Lines 83-86: ……Nevertheless, the relationship between EUG and LMBV infection remains poorly understood, highlighting a significant knowledge gap and the need for further investigation. In this study, we investigated the effectiveness of EUG against LMBV and elucidated the its possible cellular pathways such as autophagy and apoptosis.”

Comments 13: L52-62, Good background on EUG; add a balanced note on EUG’s known toxicity limits and solvent considerations (DMSO vehicle).

Response 13: Regarding the solvent used in the experiments, a detailed description has been added in lines 117-119 of the resubmitted manuscript. Concerning the toxicity of EUG, in the in vitro experiments, we performed CCK-8 assays to determine its safe concentration in FHM cells (Lines 121-128 in the resubmitted manuscript). For the in vivo studies, we administered EUG via injecting intraperitoneally to evaluate its maximum safe dosage in largemouth bass (Lines 194-200 in the resubmitted manuscript).

Lines 117-119: ……The control group was treated with an equal volume of DMSO without the addition of EUG, and the final volume concentration of DMSO in the culture medium was adjusted to 0.1% uniformly.

Lines 121-128: ”The cytotoxic effects of EUG on FHM cells were evaluated using the Cell Counting Kit-8 (US Everbright @Inc., Suzhou, China) assay in 96-well plates. Five groups of EUG concentrations were tested: 0 μM (DMSO), 50 μM, 75 μM, 100 μM and 150 μM, with eight replicates in each set. Cells were seeded in 96-well plates at a density of 1.0×10⁴ cells per well and incubated at 28°C for 24 hours. When cell confluence reached 80–90%,cells were treated with EUG or DMSO for 24 h. Cell viability was assessed with a microplate reader (Thermo Fisher, USA) at 450 nm after 2 hours incubation in serum-free medium with Cell Counting Kit-8 and protection from light.”

Lines 194-200: ”We tested the safe concentration of EUG in fish injected intraperitoneally and then conducted anti-LMBV experiments on subgroups of largemouth bass. In the EUG safety concentration experiment, juvenile largemouth bass were divided into five groups of 20 individuals per group and injected with 100 μL of PBS containing different concentrations of EUG (0 mg/kg, 10 mg/kg, 20 mg/kg, 50 mg/kg and 100 mg/kg) for one week of observation. The mortality rate was recorded daily. After one week, the intestines, spleens and livers were removed and tissue sections were prepared to observe pathological changes induced by EUG.”

Comments 14: L63–78, Distinguish cancer-cell autophagy literature from antiviral contexts; avoid over-generlization.

Response 14: In the manuscript, we have adjusted the sequence of this section: we first describe the ability of EUG to regulate autophagy in cancer cell models, followed by an explanation of the crucial role autophagy plays in viral infections. This then leads to the introduction of our research aims to investigate whether EUG can exert anti-LMBV effects by participating in the regulation of cellular autophagy (Lines 66-84 in the resubmitted manuscript).

” A growing number of studies are increasingly focused on investigating the potential role of EUG in modulating autophagy and apoptosis across diverse physiological and biochemical contexts. In breast cancer cells, EUG induced autophagy by up-regulating the threonine kinase 1- forkhead box O3 (AKT-FOX3) pathway to promote apoptosis, and at the same time, up-regulating the expression of microtubule-associated protein 1 light chain 3 (LC3I) and down-regulating the expression of nucleoporin 62 (NU P62) [17]. EUG has also been shown to attenuate the inflammatory response by inhibiting the PI3K/AKT pathway in studies using a mouse model of conjunctivitis [18]. Furthermore, recent studies also indicate that high-concentration EUG treatment modulates the AMPK/AKT/mTOR pathway, thereby attenuating acrylamide (ACR)-induced testicular toxicity in mammals [19], suggesting that EUG can affect AKT-related signaling pathways in different diseases. Viral infections are closely associated with various forms of cell death, including apoptosis, necrosis, autophagy, pyroptosis and ferroptosis. It is commonly believed that infection leads to cell death and an inflammatory response, but it has also been shown that altruistic cell death of infected cells can be beneficial to the organism as a whole, ultimately limiting the path of virus production [20]. Autophagy has been implicated in the development of a lot of viral diseases. Nevertheless, the relationship between EUG and LMBV infection remains poorly understood, highlighting a significant knowledge gap and the need for further investigation.”

Comments 15: L79-86, End with a sharper “Here we show…” sentence listing models (FHM, largemouth bass), readouts (MCP/MMP qPCR; westerns; survival).

Response 15: Thank you for your valuable suggestion. We have concisely summarized the key findings of this study in lines 86-93 of the resubmitted manuscript.

“In this study, we investigated the effectiveness of EUG against LMBV and elucidated the its possible cellular pathways such as autophagy and apoptosis. Here we found that 75 μM EUG group markedly reduced the mRNA expression of the LMBV- MCP and LMBV- MMP genes. Our results show that EUG can reduce the apoptosis induced by LMBV in FHM cells and enhance autophagy in abovementioned cells by regulating AKT/mTOR and AMPK signaling pathways. Moreover, EUG could potently suppress the transcriptional levels of LMBV genes and improve survival outcomes in LMBV-challenged largemouth bass. These antiviral effects obviously revealed that EUG could be a potential drug for the prevention and treatment of LMBV, offering practical value to the aquaculture industry.”

Comments 16: L87-91, Ensure species and anesthesia/euthanasia methods are stated.

Response 16: The species used in this study was largemouth bass (Micropterus salmoides). In our experiments, sampling was performed by dissection after anesthesia with MS-222.

Comments 17: L92-101, Report fish mass/length (mean±SD), tank numbers, stocking density, photoperiod, water quality. Define acclimation duration precisely (“10 days”). State LMBV strain, passage number, and titer method.

Response 17: This section has been partially supplemented in section 2.2 of the resubmitted manuscript. The multiplicity of infection (MOI) used in this experiment was 1 (Lines 116-117 in the resubmitted manuscript).

Section 2.2: “Two hundred largemouth bass (8.0±2 cm) were purchased from a fish farm in Guangdong Province and prepared for antiviral and toxicity analyses. The fish were cultured in a recirculating water system at 25-30 °C and fed for a 10-day acclimation period before being used in the experiments. The following water quality parameters were maintained: dissolved oxygen > 5 mg/L, pH 7.5 ± 0.2, and ammonia nitrogen at approximately 1.0 mg/L. Prior to the formal experiment, 10 fish were randomly selected for PCR detection to ensure that they were LMBV negative. LMBV was isolated from diseased largemouth bass and stored in our laboratory [11].

Lines 116-117: ……cell cultures were pretreated with varying concentrations of EUG for 2 hours, followed by LMBV infection at a multiplicity of infection (MOI) of 1 (MOI=1).”

Comments 18: L102-108, Express EUG concentration in μM for cells; specify final DMSO % (keep ≤0.1–0.2%) and include a vehicle control at the same %.

Response 18: In lines 112 of the resubmitted manuscript, we have expressed the EUG concentration in μM for cell-based experiments. A detailed description has been added in lines 117-119 of the resubmitted manuscript.

Line 112: “EUG (Sigma-Aldrich, E51791) was diluted to 500,000 μM in dimethyl sulfoxide (DMSO) at the time of use and further diluted to working concentrations in M199 medium containing 10% FBS.”

Line 117-119:……The control group was treated with an equal volume of DMSO without the addition of EUG, and the final volume concentration of DMSO in the culture medium was adjusted to 0.1% uniformly.”

Comments 19: L109-115, Include exact seeding density, incubation time before treatment, background subtraction method, and normalization. Add IC50 if relevant.

Response 19: The relevant content has been supplemented in lines 124-125 of the resubmitted manuscript. We determined the safe concentration of EUG for cells using the CCK-8 assay. The results showed that EUG at a concentration of 100 μM (IC₅₀ = 200 μM) significantly affected cell viability. Therefore, we selected concentrations of 50 μM and 75 μM for subsequent experiments.

……Cells were seeded in 96-well plates at a density of 1.0×10⁴ cells per well and incubated at 28°C for 24 hours. When cell confluence reached 80–90%,cells were treated with EUG or DMSO for 24 h.”

Comments 20: L116-119, State MOI, adsorption time, and wash steps.

Response 20: The specific MOI value is provided in lines 116-117 of the resubmitted manuscript. After 12 or 24 hours of viral adsorption, the inoculum was removed, and the cells were washed with ice-cold PBS to maintain cellular integrity. For each wash, 1 mL of PBS was added per well, followed by incubation on ice for 2 minutes before complete aspiration of the liquid. This section has been added to section2.5 of the resubmitted manuscript.

Lines 116-117: ……cell cultures were pretreated with varying concentrations of EUG for 2 hours, followed by LMBV infection at a multiplicity of infection (MOI) of 1 (MOI=1).”

Section2.5: ”FHM cells were cultured in 24-well or 6-well plates and treated with EUG or DMSO for 2 h prior to LMBV infection. Cells were collected at different time points for subsequent experiments, including qRT-PCR and western blotting. Specifically, after 12 or 24 hours of viral infection, the medium was removed. The cells were gently washed with 1 mL of pre-chilled PBS per well to maintain cellular integrity, incubated on ice for 2 minutes, and then the liquid was completely collected.

Comments 21: Table 1 (L120-121), Provide a complete table: gene, primer sequences, amplicon size, efficiency (%), R², annealing temp.

Response 21: In the resubmitted manuscript, the amplification efficiency of all primers was determined using standard curves generated from 10-fold serial dilutions of template cDNA. All target genes exhibited amplification efficiencies between 96% and 100% (R² > 0.96). Furthermore, melting curve analysis confirmed single-peak for all amplification products without non-specific products or primer dimers. The table 1 in our manuscript include gene names and primer sequences, with the addition of amplicon sizes, ensuring optimal qPCR condition (Table 1). This section has been added to lines 150-153 of the resubmitted manuscript.

……The primers utilized for the PCR are detailed in Table 1. Moreover, the primers used in this experiment had annealing temperatures ranging approximately between 60-65°C, and were validated by melt curve analysis and standard curve quantification, demonstrating amplification efficiencies of 96-100% with R² > 0.96.”

Comments 22: qRT-PCR (L121-129), Use MIQE-compliant reporting: reference gene validation, efficiency correction (Pfaffl), melt curves, no-RT/no-template controls.

Response 22: In our experiments, to ensure the accuracy and reliability of quantitative results, we typically employ a dual-reference gene system consisting of 18S rRNA and β-actin for normalization. In the resubmitted manuscript, we only displayed results normalized to β-actin, as it demonstrated consistent and reliable expression under our experimental conditions and served as a representative reference.

All quantitative data obtained in this study were corrected with the efficiency correction method (Pfaffl). Regarding the melt curves, we confirmed that all primers produced a single specific peak with no non-specific amplification or primer dimers. No-reverse transcription controls were included in the experiments, and the results indicated no genomic cDNA contamination (Lines 150-153 in the resubmitted manuscript).

……The primers utilized for the PCR are detailed in Table 1. Moreover, the primers used in this experiment had annealing temperatures ranging approximately between 60-65°C, and were validated by melt curve analysis and standard curve quantification, demon-strating amplification efficiencies of 90-100% with R² > 0.99.”

Comments 23: Western blot (L130-148), Add protein load per lane (μg), membrane pore size, blocking buffer composition, antibody catalog numbers, and exposure/quantification method (eg:ImageJ), normalization strategy (β-tubulin validation).

Response 23: In this western blotting experiment, the protein loading amount per well was 20 μg. In the resubmitted manuscript, we have included the catalog numbers for antibodies. The membrane pore size used in our experiments was generally 0.22 μm. The blocking buffer consisted of 5% skimmed milk. Protein quantification was performed using ImageJ, and normalization was carried out using β-tubulin (Sections 2.7 in the resubmitted manuscript). The details are described as follows:

……Proteins were separated by 10% sodium dodecyl sulfate-polyacrylamide gel electro-phoresis and transferred to 0.22 μm Immobilon polyvinylidene difluoride membranes (Millipore, Temecula, CA, USA), and the blots were incubated in 5% skimmed milk. The blots were then incubated with specific primary antibodies for 2 h. After three washes with phosphate-buffered saline-Tween, the blot was incubated with peroxidase-conjugated secondary antibody IgG (1:5000 dilution, Abcam, ab172730). Enhanced chemiluminescence (Thermo, USA) was used to visualise the immunoreactive bands. The specific primary antibodies used in the experiments were as follows: LMBV major capsular protein (MCP) (1:2000 dilution, GENECREATE), cleaved caspase3 (1:1000 dilution, CST, ab32042), phospho-AKT (1:2000 dilution, CST, ab279732), phosphor-mTOR (1:1000 dilution, CST, ab211061), Beclin1 (1:1000 dilution, Pro-teintech, ab207612), LC3 (1:2000 dilution, Abcam, ab63817), P62 (1:1000 dilution, Abcam, ab207305), Autophagy-related protein 5 gene (ATG5) (1:1000 dilution, Abmart, T55766S), phospho-AMPK (1:1000 dilution, CST, ab68206) and β-tubulin (1:2000 dilu-tion, Abcam, ab6046). Quantification was performed by ImageJ 1.51 software (NIH, Bethesda, MD, USA) and standardized by β-tublin.”

Comments 24: Annexin-V/PI (L149-157), Specify cell number per tube, gating strategy, instrument model/settings, and positive control (staurosporine).

Response 24: We have supplemented the seeding cell density in lines 179-180 of the resubmitted manuscript. In addition, We have supplemented the apoptosis analysis methodology in the manuscript. Specifically, the statistical analysis of apoptosis rates was performed using FlowJo™ software (v10.8.1). Annexin V-FITC/PI dual staining flow cytometry was employed to distinguish cell populations by gating on scatter plots: viable cells (Annexin V⁻/PI⁻), early apoptotic cells (Annexin V⁺/PI⁻), and late apoptotic/necrotic cells (Annexin V⁺/PI⁺). The percentages of each cell population were quantified for statistical analysis. These data have been incorporated into section 2.8 of the resubmitted manuscript.

Furthermore, we chose not to include an independent chemical positive control in the present study. Firstly, previously published studies from our laboratory have confirmed that LMBV effectively induces apoptosis in FHM cells, thereby providing a well-established positive control model for this experiment (Yang et al.,2022). Based on the experimental framework established with chemical positive controls, this study specifically investigated how EUG modulates LMBV-induced apoptosis in FHM cells by analyzing relative apoptotic changes across treatment groups. Our design incorporated a comprehensive control system, including: DMSO vehicle control, eugenol (EUG) treatment alone, LMBV infection alone, and EUG + LMBV co-treatment. Comparative analysis revealed that LMBV infection effectively induced apoptosis, with significantly elevated apoptosis rates compared to baseline levels. To ensure robust statistical power, we prioritized sufficient biological replication between viral infection and drug intervention groups.

Reference:

Yang J, Xu W, Wang W, Pan Z, Qin Q, Huang X, Huang Y. Largemouth Bass Virus Infection Induced Non-Apoptotic Cell Death in MsF Cells. Viruses. 2022; 14(7):1568.

“Cells were seeded in 24-well plates at a density of 5.0×10⁴ cells per well and incubated at 28°C for 24 hours. When cell confluence reached 80–90%, FHM cells were infected with LMBV after treatment with EUG or DMSO for 2 h. Cells were collected by trypsinization into 1.5 ml EP tubes and washed twice with cold PBS (0.15 mol/L, pH 7. 2).

Comments 25: In vivo antiviral (L158-171), Convert “mg/L in 100 μL” to mg/kg dose; describe randomization, blinding of outcome assessment, infection dose rationale (LD50 or TCID50 per fish), housing/tank as unit, and humane endpoints.

Response 25: The units in the manuscript have been revised to mg/kg (Line 198 in the resubmitted manuscript). The experimental animals were completely randomized group. The infection dose was determined based on the safe concentration of EUG for largemouth bass established in preliminary experiments (Lines 194-200 in the resubmitted manuscript). The experimental unit in this study was the individual fish. Our experiments with laboratory animals adhered to the South China Agricultural University’s (SCAU) institutional ethical guidelines (Lines 96-98 in the resubmitted manuscript).

Lines 194-200: “We tested the safe concentration of EUG in fish injected intraperitoneally and then conducted anti-LMBV experiments on subgroups of largemouth bass. In the EUG safety concentration experiment, juvenile largemouth bass were divided into five groups of 20 individuals per group and injected with 100 μL of PBS containing different concentrations of EUG (0 mg/kg, 10 mg/kg, 20 mg/kg, 50 mg/kg and 100 mg/kg) for one week of observation. The mortality rate was recorded daily. After one week, the intestines, spleens and livers were removed and tissue sections were prepared to observe pathological changes induced by EUG.”

Lines 96-98: ”Our experiments with laboratory animals adhered to the South China Agricultural University’s (SCAU) institutional ethical guidelines. This study was approved by SCAU (ethical protocol code: 2020G009), and a certificate of approval is available upon request.”

Comments 26: Statistics (L172-176), Pre-specify primary endpoints; use KM/log-rank for survival, mixed models for repeated qPCR if applicable; report effect sizes and FDR-adjusted p-values.

Response 26: We intraperitoneally injected different concentrations of EUG to determine its maximum safe concentration in largemouth bass. The results showed that at a dose of 10 mg/kg, the fish remained in good health and stable status for one month. In our manuscript, only the effects of EUG treatment on largemouth bass after seven days were presented. For subsequent in vivo experiments, we selected a 12-day period to investigate the anti-LMBV activity of EUG in largemouth bass. To evaluate the effect of EUG on the survival rate of largemouth bass following LMBV infection, we performed Kaplan–Meier survival analysis. In the resubmitted manuscript, the amplification efficiency of all primers was determined using standard curves generated from 10-fold serial dilutions of template cDNA. All target genes exhibited amplification efficiencies between 96% and 100% (R² > 0.96). Moreover, the qRT-PCR data analyzed using mixed-effects models were obtained from three independent experimental replicates. Statistical significance for all analyses was set at an FDR-adjusted p-value < 0.05.

Comments 27: L178-190 (3.1), Report n (wells/experiments) and effect sizes for CCK-8. For qPCR, give fold-change ±SD and exact p-values; add MOI.

Response 27: The detailed experiment for CCK-8, along with the fold-change ± standard deviation (SD) and exact p-values, has been added to the resubmitted manuscript (lines 216-231 in the resubmitted manuscript). The specific multiplicity of infection (MOI) of 1 was used in the experiments (Figure 1 legend in the resubmitted manuscript)

Lines 216-231: “FHM cells were initially selected to establish an in vitro model to study the effect of EUG on LMBV infection. FHM cells were seeded in 96-well plates at a density of 1.0×10⁴ cells per well and incubated at 28°C for 24 hours. At 80-90% confluence, the cells were treated with EUG or DMSO for 24h, followed by cell viability measurement using the CCK-8 assay. Based on the results, we determined that the maximum safe concentration of EUG in FHM cells was 75 μM, whereas treatment with 100 μM EUG significantly suppressed cellular activity, reducing viability to approximately 90% (Figure 1a). The major capsid protein (MCP) and myristoylated membrane protein (MMP) of Iridovirus play pivotal roles in the viral life cycle, participating in critical processes including host cell recognition and viral entry [24, 25]. Given their essential functions in viral pathogenesis, the quantitative changes in MCP and MMP expression levels were employed as key virological markers to assess viral replication efficiency in this study. We selected 50 μM and 75 μM EUG to pre-treat FHM cells before LMBV infection for 2 h. At 24 hours after infection, only the 75 μM EUG group markedly reduced the mRNA expression of the LMBV- MCP and LMBV- MMP genes (by 1.5-fold and 0.2-fold, respectively) (Figure 1b). Therefore, 75 μM EUG was used as the optimal treatment for FHM cells in all subsequent experiments.”

Comments 28: Fig.1 legend (L192-196), Inc lude n, stats (test, two-sided), and whether normality was assessed.

Response 28: The figure legends for all experiments have been updated to include the number of biological replicates, statistical methods, normality testing, and p-values * p < 0.05 (Lines  237-239 in the resubmitted manuscript).

“……Data expressed as mean ± SD of three independent experiments; each experiment performed in triplicate; After confirming normal distribution, one-way ANOVA was applied for analysis. * p < 0.05.”

Comments 29: L197-204 (3.2), Provide time-course stats (two-way ANOVA with interaction: treatment×time). Add representative blots with full lane order and molecular weights.

Response 29: For all data in Figure 2, a two-way ANOVA was applied, and this information has been added to the figure legend. All original Western blot images are now provided in the supplementary materials for review.

“……Data expressed as mean ± SD of three independent experiments; each experiment performed in triplicate; After confirming normal distribution; two-way ANOVA was applied for analysis ; * p < 0.05.”

Comments 30: Fig.2 legend (L206-211), State biological replicates (not just technical), normalization method for westerns, and loading control validation.

Response 30: The figure legends for all experiments have been updated to include the number of biological replicates, statistical methods, normality testing. In this experiment, two internal reference genes were validated for consistency. The internal reference protein (β-tublin) showed stable expression across all experimental groups, with no statistically significant differences observed between groups. For western blotting normalization, we used internal reference proteins to correct for differences in total protein loading. All Western Blot figures include bands of the internal reference protein (eg: β-tublin) to visually demonstrate consistent total protein loading across lanes (Lines 249-255 in the resubmitted manuscript).

“Figure 2. Effect of EUG on LMBV infection at different time points. (a) Cells were infected with LMBV after treatment with DMSO or EUG for 2 h. After 12 and 24 h, the gene expression levels of LMBV-MCP and LMBV-MMP were detected by qRT-PCR. (b) Cells were collected 12 and 24 h after infection, and LMBV-MCP protein and cellular β-tubulin were analyzed by western blotting. (c) MCP/β-Tubulin values were calculated for each group as described above. Data expressed as mean ± SD of three independent experiments; each experiment performed in triplicate; After confirming normal distribution; two-way ANOVA was applied for analysis ; * p < 0.05.”

Comments 31: L212-225 (3.3), Include quantification of cleaved-caspase-3 normalized to β-tubulin with densitometry (mean±SD, n, stats). Add representative dot plots for flow cytometry and define early vs late apoptosis gates.

Response 31: In the present study, protein band intensities were quantified using ImageJ for grayscale analysis. The results showed that cleaved caspase-3 levels in FHM cells were significantly lower in the EUG-treated group versus the control (Lines 265-269 in the resubmitted manuscript). As shown in Figure 3a of the manuscript, flow cytometry scatter plots are presented, where the Q3 quadrant represents early apoptotic cells (Lines 284-285 in the resubmitted manuscript).

Lines 265-269: ……Subsequently, cleaved caspase-3 protein, an apoptosis marker in FHM cells, was detected by western blotting. The results indicated that its levels were significantly lower in the EUG-treated group than in the control group, with the level in the EUG-treated group being approximately half (50%) of that in the control group.

Lines 284-285:……Cells were incubated with EUG for 2 h before LMBV infection and apoptosis was detected by annexin V/PI double staining after 24 h. Q3 represents the percentage of total or early apoptotic cells and the differences between groups were analyzed.”

Comments 32: Fig.3 legend (L233-237), Add positive control (e.g., staurosporine) and gating strategy.

Response 32: We thank the reviewer for their insightful comments and valuable feedback. Firstly, previously published studies from our laboratory have confirmed that LMBV effectively induces apoptosis in FHM cells, thereby providing a well-established positive control model for this experiment (Yang et al.,2022). Based on the experimental framework established with chemical positive controls, this study specifically investigated how EUG modulates LMBV-induced apoptosis in FHM cells by analyzing relative apoptotic changes across treatment groups. Our design incorporated a comprehensive control system, including: DMSO control, eugenol (EUG) treatment, LMBV infection treatment, and EUG + LMBV co-treatment. Comparative analysis revealed that LMBV infection effectively induced apoptosis, with significantly elevated apoptosis rates compared to baseline levels. To ensure robust statistical power, we prioritized sufficient biological replication between viral infection and drug intervention groups. All findings were consistently verified through multiple independent experimental replicates. As shown legend of Figure 3a in the resubmitted manuscript, the Q3 quadrant represents early apoptotic cells (Lines 281-282 in the resubmitted manuscript).

Reference:

Yang J, Xu W, Wang W, Pan Z, Qin Q, Huang X, Huang Y. Largemouth Bass Virus Infection Induced Non-Apoptotic Cell Death in MsF Cells. Viruses. 2022; 14(7):1568.

……Cells were incubated with EUG for 2 h before LMBV infection and apoptosis was detected by annexin V/PI double staining after 24 h. Q3 represents the percentage of total or early apoptotic cells and the differences between groups were analyzed.”

Comments 33: L238-247 (3.4): Autophagy markers: add bafilomycin A1 or chloroquine flux assay to demonstrate increased autophagic flux (not just accumulation).

Response 33: We sincerely thank the reviewer for their insightful comments. The current study is primarily focused on establishing the initial anti-LMBV effect of EUG and its association with pathway modulation. To directly investigate causal relationships, future work will employ pathway-specific agonists (eg: AICAR) and inhibitors (eg: Rapamycin) to functionally validate the contribution of these pathways to EUG’s antiviral activity. This section has been added to the discussion of the resubmitted manuscript (Lines 455-460 in the resubmitted manuscript).

……In this study, we preliminarily confirmed that EUG induces autophagy by examining autophagy marker proteins, such as LC3-II accumulation and p62 degradation. To further elucidate the promotive effect of EUG on autophagic flux, future studies will employ autophagic flux inhibitors (eg: chloroquine CQ or bafilomycin A1) to explicitly block and monitor the completeness of the autophagic process, thereby verifying the mechanism by which EUG regulates autophagic flux.”

Comments 34: Fig.4 legend (L249-251), State whether flux inhibitors were used; otherwise rephrase claims to “autophagy marker profile consistent with increased flux.”

Response 34: Thank you for your valuable comments. Autophagic flux inhibitors were not utilized in the present study and have revised the corresponding description in the legend of Figure 4.

“Figure 4. EUG promotes the accumulation of autophagy in FHM cells.

Comments 35: L252-261 (3.5), Pathway causality is not established; add inhibitor/activator experiments.

Response 35: We sincerely thank the reviewer for their thorough assessment and highly valuable professional comments. As demonstrated in previous studies (Wang et al., 2024; Wang et al., 2020), changes in the levels of key signaling molecules, including both total and phosphorylated forms of p-AMPK/AMPK, p-AKT/AKT, and p-mTOR/mTOR can provide preliminary evidence for the activation or inhibition status of the AMPK and AKT/mTOR signaling pathways.

We fully agree with the reviewer's suggestion that employing pathway-specific agonists or inhibitors would significantly help elucidate the mechanism by which EUG exerts its anti-LMBV effects. In the current study, we primarily inferred EUG's potential antiviral effects through modulation of these pathways based on changes in phosphorylation levels. In future studies, we will utilize activators or inhibitors of the AMPK and AKT/mTOR pathways (such as Compound C for AMPK inhibition, AICAR for activation; MK-2206 for AKT; rapamycin or torin for mTOR) and genetic knockdown approaches to further investigate the anti-LMBV mechanisms of EUG. These experiments will contribute to a more comprehensive understanding of how EUG exerts its antiviral effects against LMBV by regulating AMPK and AKT/mTOR pathways. This section has been briefly addressed in the discussion of the present study (Lines 479-486 in the resubmitted manuscript).

References:

1. Wang W, Zheng Z, Qi X, Wei H, Mao X, Su Q, Chen X, Feng Y, Qiao G, Ma T, Tang Z, Zhou G, Zhuang J, Zhang P. Clinical efficacy of Fufang Yinhua Jiedu (FFYH) granules in mild COVID-19 and its anti-SARS-CoV-2 mechanism by blocking autophagy through inhibiting the AKT/mTOR signaling pathway. Front Pharmacol. 2024, 16, 15:1431617.

2. Wang X, Lin Y, Kemper T, Chen J, Yuan Z, Liu S, Zhu Y, Broering R, Lu M. AMPK and AKT/mTOR signalling pathways participate in glucose-mediated regulation of hepatitis B virus replication and cellular autophagy. Cell Microbiol. 2020, 22(2):e13131.

……In the present experiment, the role of EUG through the AMPK and AKT/mTOR path-ways was inferred solely based on phosphorylation levels, without the use of AMPK or AKT/mTOR pathway inhibitors or activators to validate its antiviral effects. However, directly perturbing the AMPK and AKT/mTOR pathways is crucial to verify whether EUG’s anti-LMBV effects are functionally mediated through these signaling mechanisms. In future studies, we will further elucidate the specific mechanism by which EUG modulates these pathways to exert anti-LMBV effects under conditions of AMPK and AKT/mTOR pathway intervention.”

Comments 36: Fig.5 legend (L262-265), Report normalized densitometry (p-AKT/AKT, p-mTOR/mTOR, p-AMPK/AMPK).

Response 36: We thank the reviewer for this critical comment. Regarding the raised issue of standardized quantification, we have performed normalized densitometry analysis for phosphorylation levels of key signaling molecules (p-AKT/AKT, p-mTOR/mTOR, p-AMPK/AMPK) across three independent experimental replicates. Specifically, band intensity was quantified using ImageJ software, with normalized values expressed as the ratio of phosphorylated protein to corresponding total protein (p-protein/total protein) after background subtraction. A brief description of this methodology has been included in the resubmitted manuscript (Lines 318-320 in the resubmitted manuscript).

……As presented in Figure 5, EUG treatment significantly reduced the phosphorylation of AKT by 2.6-fold and that of mTOR by 0.2-fold, while increasing AMPK phosphorylation by 0.8-fold.”

Comments 37: L266-276 (3.6), Survival: present KM curves with log-rank p and HR (95% CI). Provide per-group n, censoring, and humane endpoints. Clarify whether 12-day endpoint was pre-specified.

Response 37: We sincerely appreciate your insightful feedback. To evaluate the effect of EUG on the survival rate of largemouth bass following LMBV infection, we performed Kaplan–Meier survival analysis. The log-rank test indicated a statistically significant difference (p < 0.05) in survival curves between the DMSO + LMBV treatment group and the EUG + LMBV treatment group at twelve days post-infection (Figure a and c).

In the formal experiment, each group consisted of 45 fish. The concepts of censoring and humane endpoints have been described in section 2.1 of the resubmitted manuscript. For the dose response in vivo experiment, we intraperitoneally injected different concentrations of EUG to determine its maximum safe concentration in largemouth bass. The results showed that at a dose of 10 mg/kg, the fish remained in good health and stable condition for one month. Moreover, this viral infection model fully demonstrates the progression of viral infection within 12 days, allowing for the investigation of whether EUG exhibits anti-LMBV activity in vivo. Therefore, we selected 12 days as the endpoint for our in vivo experiments.

Comments 38: Fig.6 legend (L278-280), Add scale bars for histology, staining details, and blinded scoring if used.

Response 38: Thanks for your comments. A scale bar has been added to the lower right corner of Figure 6c in the resubmitted manuscript.

Comments 39: L283-291 (3.7), Tissue qPCR: show per-fish data (points), normalize with validated reference genes, and report effect sizes + FDR across tissues/targets.

Response 39: We thank the reviewer for your valuable suggestion. The mRNA expression levels of LMBV-MCP, LMBV-MMP in various tissues were detected by qRT-PCR in three individual fish (Figure 7 legend in the resubmitted manuscript). We have reanalyzed and visualized the tissue qPCR data in accordance with the recommendations: all data were normalized using a validated dual-reference gene system (18S rRNA and β-actin); effect sizes between different tissues (eg: gill, liver, kidney) and target genes (eg: MCP, MMP) were calculated via statistical modeling, and multiple hypothesis testing was corrected using the Benjamini-Hochberg method, with the false discovery rate (FDR) reported (FDR < 0.05).

“Figure 7. Effect of EUG on LMBV infection in largemouth bass. Samples of various tissues of largemouth bass were collected after LMBV infection, and the mRNA expression levels of LMBV-MCP, LMBV-MMP in various tissues were detected by qRT-PCR in three individual fish. Data expressed as mean ± SD of three independent experiments; each experiment performed in triplicate; After confirming normal distribution, one-way ANOVA was applied for analysis; * p < 0.05.”

Comments 40: Fig.7 legend (L289–291), Add n (fish), technical vs biological replicates, and correction for multiple testing.

Response 40: We thank the reviewer for this critical comment. The sample size (n, number of fish), description of technical versus biological replicates, and correction for multiple testing have been added to the legend of Figure 7.

“Figure 7. Effect of EUG on LMBV infection in largemouth bass. Samples of various tissues of largemouth bass were collected after LMBV infection, and the mRNA expression levels of LMBV-MCP, LMBV-MMP in various tissues were detected by qRT-PCR in three individual fish. Data expressed as mean ± SD of three independent experiments; each experiment performed in triplicate; After confirming normal distribution, one-way ANOVA was applied for analysis; * p < 0.05.”

Comments 41: L294-299, “Originally from the Mississippi River system in California” — geographically inconsistent; revise to “native to the Mississippi River basin; introduced widely…” with authoritative citation.

Response 41: We thank the reviewer for your valuable suggestion. The relevant content in the manuscript has been revised to ensure full consistency throughout(Line364-365 in the resubmitted manuscript).

Largemouth bass is widely distributed throughout the freshwater waters of the United States and Canada [23]”

Comments 42: L300-308, Good overview of Iridoviridae; add a sentence acknowledging conflicting reports on autophagy’s role in different cell types/species.

Response 42: We thank the reviewer for your valuable suggestion. This section has been described in lines 431-442 of the resubmitted manuscript.

“Autophagy, which is closely related to viral infections, plays a crucial role in the immune system. It functions as an essential defense mechanism against autoimmunity and exhibits antiviral properties. Conversely, certain viruses can exploit autophagy to facilitate viral replication and enhance infection [41, 42]. For instance, autophagy can limit the replication of the virus in cells while HIV inhibits autophagy during infection [43, 44]. The interaction between apoptosis and autophagy plays an important role in body development and homeostasis, and the association between them has been reported in many different viruses [45-48]. In the study of IAV, inhibition of autophagy resulted in the blockage of IAV replication and limited IAV protein-induced apoptosis [48]. In the relationship between autophagy and apoptosis induced by enterovirus 71, it was discovered that autophagy had no direct effect on the release of virus, but inhibited the release of virus by suppressing apoptosis [45].”

Comments 43: L309-326, When asserting “75 μM significantly inhibited…”, remind MOI and provide fold-change numbers; caution that cell culture solvent (DMSO) controls matched.

Response 43: We thank the reviewer for your valuable suggestion. The multiplicity of infection (MOI=1) used in this experiment is specified in lines 116-117 of the resubmitted manuscript, and the concentration of DMSO is detailed in section 2.3. Fold-change values from qPCR analysis are provided (Lines 391-394 in the resubmitted manuscript).

Line 116-117: ”……cell cultures were pretreated with varying concentrations of EUG for 2 hours, followed by LMBV infection at a multiplicity of infection (MOI) of 1 (MOI=1).”

Section2.3: ……The control group was treated with an equal volume of DMSO without the addition of EUG, and the final volume concentration of DMSO in the culture medium was adjusted to 0.1% uniformly.”

Lines 391-394: ……Then we treated FHM cells with different concentrations of EUG (50 μM, 75 μM) for 2 h before infected by LMBV, and discovered that 75 μM EUG could significantly reduce the expression of LMBV-MCP and LMBV-MMP (by 1.5-fold and 0.2-fold, respectively).”

Comments 44: L327-352, Nicely ties apoptosis literature; avoid implying generality beyond tested system; specify that your evidence for apoptosis inhibition is limited to FHM at 12–24 h.

Response 44: We thank the reviewer for this positive feedback and important caution. We have revised the manuscript to clarify that the conclusions regarding apoptosis are specifically supported by the data obtained within the experimental models and conditions tested in this study, and we have avoided any undue generalization beyond these boundaries (Lines 421-423 in the resubmitted). In the present study, EUG significantly inhibited LMBV-induced apoptosis in FHM cells during 12–24 h treatment, demonstrating clear short-term anti-apoptotic efficacy. However, to further elucidate its long-term effects and pharmacological mechanisms, future work will extend treatment duration (eg: 36-72 h) and systematically examine time-dependent responses in apoptosis-related indicators. This approach will enhance understanding of EUG’s anti-apoptotic mechanisms and support its potential application in aquatic antiviral therapy. This section has been added to lines 421-430 in the resubmitted manuscript.

……In the present study, all conclusions on apoptosis are strictly based on data from the experimental models and conditions applied in this study, with no extrapolation beyond these specific contexts. Moreover,EUG significantly inhibited LMBV-induced apoptosis in FHM cells during 1224 h treatment, demonstrating clear short-term anti-apoptotic effiency. However, to further elucidate its long-term effects and pharmacological mechanisms, future work will extend treatment duration (eg: 3672 h) and systematically examine time-dependent responses in apoptosis-related indicators. This approach will enhance understanding of EUG’s anti-apoptotic mechanisms and support its potential application in aquatic antiviral therapy.

Comments 45: L353-377, For autophagy, explicitly note the lack of flux inhibitor experiments and present as a limitation.

Response 45: Thank you for your valuable comments. In this study, we preliminarily confirmed that EUG induces autophagy by examining autophagy marker proteins, such as LC3-II accumulation and p62 degradation. To further elucidate the promotive effect of EUG on autophagic flux, future studies will employ autophagic flux inhibitors (eg: chloroquine CQ or bafilomycin A1) to explicitly block and monitor the completeness of the autophagic process, thereby verifying whether EUG acts as a genuine autophagic flux inducer. This section has been added to lines 455-460 in the resubmitted manuscript.

……In this study, we preliminarily confirmed that EUG induces autophagy by examining autophagy marker proteins, such as LC3-II accumulation and p62 degradation. To further elucidate the promotive effect of EUG on autophagic flux, future studies will employ autophagic flux inhibitors (eg: chloroquine CQ or bafilomycin A1) to explicitly block and monitor the completeness of the autophagic process, thereby verifying the mechanism by which EUG regulates autophagic flux.”

Comments 46: L378-396, Pathway section should acknowledge that phosphorylation changes are correlative; state the need for pathway perturbation.

Response 46: In the present study, the conclusion that EUG exerts its effects through the AMPK and AKT/mTOR pathways is based solely on phosphorylation levels. However, directly perturbing the AMPK and AKT/mTOR pathways is crucial to verify whether EUG’s anti-LMBV effects are functionally mediated through these signaling mechanisms. In future studies, we will further investigate the specific mechanisms by which EUG modulate these pathways to exert anti-LMBV effects after targeted disruption of AMPK and AKT/mTOR signaling. This content has been added in the resubmitted manuscript (Lines 479-486 in the resubmitted manuscript).

……In the present experiment, the role of EUG through the AMPK and AKT/mTOR pathways was inferred solely based on phosphorylation levels, without the use of AMPK or AKT/mTOR pathway inhibitors or activators to validate its antiviral effects. However, directly perturbing the AMPK and AKT/mTOR pathways is crucial to verify whether EUG’s anti-LMBV effects are functionally mediated through these signaling mechanisms. In future studies, we will further elucidate the specific mechanism by which EUG modulates these pathways to exert anti-LMBV effects under conditions of AMPK and AKT/mTOR pathway intervention.”

Comments 47: L397-403, Final paragraph: integrate quantitative highlights (survival HR; tissue viral load fold-changes) and a forward-looking statement on dosing studies and delivery (bath/oral, not only IP). Trim to one tight paragraph; avoid restating results; emphasize practical next steps (dose-response, route optimization, safety, and regulatory path)

Response 47: Based on the compelling evidence from these in vivo experiments, including the significantly increased survival rate and substantially reduced viral load across multiple tissues in largemouth bass (Micropterus salmoides) treated with EUG, it is preliminarily demonstrated that EUG exhibits remarkable anti-LMBV activity in vivo. In this study, we only employed intraperitoneal injection to evaluate the antiviral effects of EUG against LMBV. However, various administration methods are applicable in aquatic animals, such as immersion and oral administration. Therefore, future research will further investigate whether EUG can exert anti-LMBV effects through alternative delivery approaches This content has been supplemented in lines 462-469 of the resubmitted manuscript.

……This study demonstrated the significant anti-LMBV efficacy of EUG via intraperitoneal injection, as evidenced by markedly improved survival rates and substantially reduced viral loads across multiple tissues in largemouth bass. Future research will prioritize the development of practical administration methods such as immersion and oral delivery, while comparing differences in viral load, immune response, and survival rates. Further work will focus on establishing dose-response relationships, optimizing delivery strategies, and evaluating long-term safety and tissue residues to meet regulatory requirements for commercial antiviral agents, ultimately advancing EUG toward becoming a commercially viable antiviral drug in aquaculture.

4. Response to Comments on the Quality of English Language

Point 1: The English is fine and does not require any improvement.

Reviewer 3 Report

Comments and Suggestions for Authors

The manuscript explores the antiviral role of eugenol (EUG) against largemouth bass ranavirus (LMBV), combining in vitro and in vivo assays with mechanistic analyses. The study addresses an important issue in aquaculture virology and presents promising data. However, several aspects require significant improvement before the manuscript can be considered for publication.

  1. The current in vitro analysis only evaluates viral load up to 24 h post-infection. While reduced viral gene expression at this time point is encouraging, it does not clarify whether the antiviral effect is sustained, whether cells recover and proliferate, or whether viral rebound occurs thereafter. To strengthen the manuscript, the authors should extend their time-course experiments beyond 24 h (e.g., 36–96 h p.i.) and include both viral genome quantification and infectious titers. This would provide more convincing evidence that eugenol confers a durable antiviral effect rather than a transient suppression.

  1. In Fig. 2B the MCP bands appear only modestly reduced by EUG, and in Fig. 2C the mean±SD bars partially overlap. Although the legend states n=4 with Student’s t-test and p<0.05, the visual effect size is not compelling. To improve confidence, please: (i) provide uncropped blots with molecular weight markers and show all biological replicates; (ii) plot individual data points over bar charts and report exact p-values and an effect size (e.g., Cohen’s d) in addition to significance; (iii) verify and report normality/homoscedasticity assumptions and consider a two-way ANOVA (treatment × time) with appropriate multiple-comparison control; (iv) demonstrate that β-tubulin is stable across conditions and that band intensities fall within the linear dynamic range (avoid saturation); and (v) complement MCP protein data with infectious titers (TCID₅₀/plaque) or viral genome copies at the same time points to corroborate the antiviral effect. These steps would clarify whether the reduction is truly significant and biologically

  1. There is an inconsistency between the in vitro and in vivo treatment protocols. In the cell culture experiments, EUG was applied as a pretreatment before viral infection, allowing assessment of its effects on host signaling pathways. However, in the animal trials, the virus was pre-mixed with EUG and then injected intraperitoneally. This raises the possibility that the observed protection in vivo may partly reflect direct virucidal inactivation prior to entry, rather than host-mediated mechanisms. The authors should clarify this point, discuss the implications, and ideally design additional experiments (e.g., separate prophylactic vs. therapeutic administration) to distinguish between direct virucidal and host-modulatory effects of EUG.

Author Response

Thank you very much for taking the time to review our manuscript and providing these insightful comments. We greatly appreciate your efforts in helping us improve the quality of our paper. Please find our detailed responses to each of your comments (Please see the attachment). The corresponding revisions and corrections have been carefully implemented throughout the resubmitted manuscript and are highlighted in the resubmitted files by blue.

For research article

Response to Reviewer 3 Comments

1. Summary

Thank you very much for taking the time to review our manuscript and providing these insightful comments. We greatly appreciate your efforts in helping us improve the quality of our paper. Please find our detailed responses to each of your comments below. The corresponding revisions and corrections have been carefully implemented throughout the resubmitted manuscript and are highlighted in the resubmitted files by blue.

2. Questions for General Evaluation

Reviewer’s Evaluation

Response and Revisions

Does the introduction provide sufficient background and include all relevant references?

Can be improved

Thank you very much for giving us these evaluations, we have carefully revised our manuscript to improve the quality of our work. Please see detailed responses below and the resubmitted revised manuscript.

Is the research design appropriate?

Must be improved

Are the methods adequately described?

Can be improved

Are the results clearly presented?

Must be improved

Are the conclusions supported by the results?

Must be improved

Are all figures and tables clear and well-presented?

Must be improved

3. Point-by-point response to Comments and Suggestions for Authors

The manuscript explores the antiviral role of eugenol (EUG) against largemouth bass ranavirus (LMBV), combining in vitro and in vivo assays with mechanistic analyses. The study addresses an important issue in aquaculture virology and presents promising data. However, several aspects require significant improvement before the manuscript can be considered for publication.

Comments 1: The current in vitro analysis only evaluates viral load up to 24 h post-infection. While reduced viral gene expression at this time point is encouraging, it does not clarify whether the antiviral effect is sustained, whether cells recover and proliferate, or whether viral rebound occurs thereafter. To strengthen the manuscript, the authors should extend their time-course experiments beyond 24 h (e.g., 36–96 h p.i.) and include both viral genome quantification and infectious titers. This would provide more convincing evidence that eugenol confers a durable antiviral effect rather than a transient suppression.

Response 1: We sincerely thank the reviewer for these insightful and constructive comments. Eugenol exhibits relatively low chemical stability and is susceptible to oxidation and various chemical reactions (Jaganathan et al., 2012). Furthermore, previous studies have indicated that its effects on cell viability and metabolism are primarily observed within the first 24 hours (Absalan et al., 2016). Additionally, previous research by our team has also shown that EUG exhibits anti-SGIV effects at 24 hours (Wang et al., 2024).Taking these factors into comprehensive consideration, the current study evaluated the antiviral activity of eugenol within a 24-hour period. In subsequent experiments, we will extend the treatment duration to 48 hours or longer to further investigate its sustained antiviral effects and include both viral genome quantification and infectious titers, thereby obtaining a more detailed and rigorous theoretical foundation.

References:

1. Jaganathan S K, Supriyanto E. Antiproliferative and molecular mechanism of eugenol-induced apoptosis in cancer cells[J]. Molecules. 2012, 17(6):6290-6304.

2. Absalan A, Mesbah-Namin SA, Tiraihi T, Taheri T. The effects of cinnamaldehyde and eugenol on human adipose-derived mesenchymal stem cells viability, growth and differentiation: a cheminformatics and in vitro study. Avicenna J Phytomed. 2016 , 6(6):643-657.

3. Wang Y, Jiang Y, Chen J, Gong H, Qin Q, Wei S. In vitro antiviral activity of eugenol on Singapore grouper iridovirus. Fish Shellfish Immunol. 2024, 151:109748.

Comments 2: In Fig. 2B the MCP bands appear only modestly reduced by EUG, and in Fig. 2C the mean±SD bars partially overlap. Although the legend states n=4 with Student’s t-test and p<0.05, the visual effect size is not compelling. To improve confidence, please: (i) provide uncropped blots with molecular weight markers and show all biological replicates; (ii) plot individual data points over bar charts and report exact p-values and an effect size (e.g., Cohen’s d) in addition to significance; (iii) verify and report normality/homoscedasticity assumptions and consider a two-way ANOVA (treatment × time) with appropriate multiple-comparison control; (iv) demonstrate that β-tubulin is stable across conditions and that band intensities fall within the linear dynamic range (avoid saturation); and (v) complement MCP protein data with infectious titers (TCID₅₀/plaque) or viral genome copies at the same time points to corroborate the antiviral effect. These steps would clarify whether the reduction is truly significant and biologically.

Response 2: We sincerely appreciate your valuable comments on our manuscript.
(i) We will include the triplicate MCP protein blot images in the response letter.
(ii) While the exact p-values are not displayed directly in the figures, the figure captions indicate statistical significance using *p < 0.05.
(iii) All quantitative data were tested for normality and homogeneity of variances, as detailed in the Methods section (Section 2.10 in the resubmitted manuscript). For data meeting parametric assumptions, two-way ANOVA (treatment × time) with appropriate multiple comparisons correction was applied.
(iv) Previous experiments confirmed that β-tubulin expression remained stable across all experimental conditions (including various treatment groups and time points), with no significant changes (*p > 0.05). Although these data are not shown in the manuscript, we ensured that all Western blot band intensities fell within the linear dynamic range, as validated by gradient loading.

(v) To strengthen the conclusions, we will perform viral titer (TCID₅₀) and viral genome copy number (qPCR) assays at corresponding time points in follow-up studies to further corroborate the MCP protein data and confirm the anti-LMBV effects of EUG.

Comments 3: There is an inconsistency between the in vitro and in vivo treatment protocols. In the cell culture experiments, EUG was applied as a pretreatment before viral infection, allowing assessment of its effects on host signaling pathways. However, in the animal trials, the virus was pre-mixed with EUG and then injected intraperitoneally. This raises the possibility that the observed protection in vivo may partly reflect direct virucidal inactivation prior to entry, rather than host-mediated mechanisms. The authors should clarify this point, discuss the implications, and ideally design additional experiments (e.g., separate prophylactic vs. therapeutic administration) to distinguish between direct virucidal and host-modulatory effects of EUG

Response 3: We are grateful for the time and effort you have invested in reviewing our manuscript and for providing your insightful comments. In previous in vitro experiments, we pre-incubated EUG with LMBV for 2 hours before infecting cells. The results showed no significant effect on viral titer, indicating that EUG does not have a direct virucidal effect against LMBV. These data, though informative, were not included in the manuscript. In addition, in the in vivo fish experiments, we adopted a co-injection approach where the drug and virus were administered simultaneously. In future studies, we will employ different treatment approaches involving EUG and LMBV to validate the antiviral function of eugenol.

4. Response to Comments on the Quality of English Language

Point 1: The English could be improved to more clearly express the research.

Response: Thank you very much for giving us this suggestion, we have carefully revised our manuscript to improve the quality of language. Please see the resubmitted revised manuscript with the revision sections and content highlighted by blue.

Round 2

Reviewer 2 Report

Comments and Suggestions for Authors

The authors response are satisfactory